# FedDA: Faster Adaptive Gradient Methods for Federated Constrained Optimization

**Junyi Li**
Department of Computer Science
University of Maryland College Park
College Park, MD 20742
junyili.ai@gmail.com

**Feihu Huang**
Electrical and Computer Engineering
University of Pittsburgh
Pittsburgh, PA 15261
huangfeihu2018@gmail.com

**Heng Huang**[*]
Department of Computer Science
University of Maryland College Park
College Park, MD 20742
henghuanghh@gmail.com

## Abstract

Federated learning (FL) is an emerging learning paradigm where a set of distributed clients learns a task under the coordination of a server. The FedAvg algorithm is one of the most widely used methods in FL. In FedAvg, the learning rate is a constant rather than changing adaptively. Adaptive gradient methods have demonstrated superior performance over the constant learning rate schedules in non-distributed settings, and they have recently been adapted to FL. However, the majority of these methods are designed for unconstrained settings. Meanwhile, many crucial FL applications, like disease diagnosis and biomarker identification, often rely on constrained formulations such as Lasso and group Lasso. It remains an open question as to whether adaptive gradient methods can be effectively applied to FL problems with constrains. In this work, we introduce **FedDA**, a novel adaptive gradient framework for FL. This framework utilizes a restarted dual averaging technique and is compatible with a range of gradient estimation methods and adaptive learning rate schedules. Specifically, an instantiation of our framework FedDA-MVR achieves sample complexity $\tilde{O}(K^{-1}\epsilon^{-1.5})$ and communication complexity $\tilde{O}(K^{-0.25}\epsilon^{-1.25})$ for finding a stationary point $\epsilon$ in the constrained setting with $K$ be the number of clients. We conduct experiments over both constrained and unconstrained tasks to confirm the effectiveness of our approach.

## 1 Introduction

As an emerging machine learning technique, federated learning (FL) has recently been applied to many important health and biomedicine applications (Moshawrab et al., 2023; Joshi et al., 2022; Antunes et al., 2022; Xu et al., 2021a; Rahman et al., 2022). The FL enables big data analyses in healthcare (especially for rare diseases) via allowing a set of distributed located hospitals, medical centers, insurance companies, *etc.*, to jointly perform a machine learning task under the coordination of a central server over their privately-held data. For example, in a recent FL study, the data from 71 sites across 6 continents are used to generate an automatic tumor boundary detector for the rare disease of glioblastoma (Pati et al., 2022). Among these FL healthcare applications, federated biomarker identification (Sheller et al., 2020) is one of the most important learning tasks to help researchers and clinicians detect informative biomarkers from distributed biomedical datasets to understand the underlying disease mechanisms, diagnose disease earlier, and design drugs. Different to prediction tasks, the federated biomarker identification often utilizes the constrained formulations, such as Lasso and group Lasso, which often requires specific optimization solvers.

---
[*]This work was partially supported by NSF IIS 2347592, 2347604, 2348159, 2348169, DBI 2405416, CCF 2348306, CNS 2347617.

Table 1: **Comparisons of representative Federated Learning algorithms for finding an $\epsilon$-stationary point of equation 1** *i.e.,* $\|\nabla f(x)\|^2 \leq \epsilon$ or its equivalent variants. $Gc(f, \epsilon)$ denotes the number of gradient queries *w.r.t.* $f^{(k)}(x)$ for $k \in [K]$; $Cc(f, \epsilon)$ denotes the number of communication rounds; **State** means what state the algorithm maintains locally (Primal/Dual); **Local-Adaptive** means whether the algorithm performs adaptive gradient descent locally or not; **Constrained** means whether the algorithm can solve both constrained and unconstrained problems or not.

| Type | Algorithm | $Gc(f, \epsilon)$ | $Cc(f, \epsilon)$ | State | Local-Adaptive | Constrained |
|---|---|---|---|---|---|---|
| Non-adaptive | FedAvg McMahan et al. (2017) | $O(K^{-1}\epsilon^{-2})$ | $O(\epsilon^{-1.5})$ | Primal/Dual | $\times$ | $\times$ |
| | FedDualAvg Yuan et al. (2021) | $O(K^{-1}\epsilon^{-2})$ | $O(K^{-1}\epsilon^{-2})$ | Dual | $\times$ | $\checkmark$ |
| | FedCM Xu et al. (2021b) | $\tilde{O}(K^{-1}\epsilon^{-2})$ | $\tilde{O}(K^{-1}\epsilon^{-2})$ | Primal/Dual | $\times$ | $\times$ |
| | FedGLOMO Das et al. (2020) | $\tilde{O}(K^{-0.5}\epsilon^{-1.5})$ | $\tilde{O}(\epsilon^{-1.5})$ | Primal/Dual | $\times$ | $\times$ |
| | STEM Khanduri et al. (2021a) | $\tilde{O}(K^{-1}\epsilon^{-1.5})$ | $\tilde{O}(\epsilon^{-1})$ | Primal/Dual | $\times$ | $\times$ |
| Adaptive | FedAdam Reddi et al. (2020) | $O(K^{-1}\epsilon^{-2})$ | $O(K^{-0.5}\epsilon^{-1})$ | Primal | $\times$ | $\times$ |
| | Local-AMSGrad Chen et al. (2020b) | $O(K^{-1}\epsilon^{-2})$ | $O(K^{-1}\epsilon^{-2})$ | Primal | $\checkmark$ | $\times$ |
| | MIME-MVR Karimireddy et al. (2020a) | $\tilde{O}(K^{-0.5}\epsilon^{-1.5})$ | $O(K^{-0.5}\epsilon^{-1.5})$ | Primal | $\checkmark$ | $\times$ |
| | FAFED Wu et al. (2022) | $\tilde{O}(K^{-1}\epsilon^{-1.5})$ | $\tilde{O}(\epsilon^{-1})$ | Primal | $\checkmark$ | $\times$ |
| | **FedDA-MVR(Ours)** | $\tilde{O}(K^{-1}\epsilon^{-1.5})$ | $\tilde{O}(K^{-0.25}\epsilon^{-1.25})$ | Dual | $\checkmark$ | $\checkmark$ |

A widely used method in Federated Learning (FL) is the FedAvg (Local-SGD) algorithm (McMahan et al., 2017). As indicated by its name, FedAvg performs (stochastic) gradient descent steps on each client and averages local states periodically. Recently, researchers incorporated adaptive gradient methods to the FL setting (Reddi et al., 2020; Karimireddy et al., 2020a; Chen et al., 2020b) to accelerate FedAvg. For instance, Reddi et al. (2016) introduced FedAdam, which incorporates the Adam optimizer for server updates; Karimireddy et al. (2020a) proposed MIME, which supports adaptive gradients in local updates. These adaptive gradient methods have demonstrated significant performance enhancements compared to FedAvg. However, these methods are designed for the unconstrained setting and cannot be directly applied over the constrained formulations as in the biomarker identification tasks. In fact, FL problems with constraints remain under-explored in the literature. In (Yuan et al., 2021), authors proposed FedDualAvg to solve FL problems with non-smooth regularizers. In FedDualAvg, model parameters are transformed into a dual space using a map determined by the regularizer. Averaging is then conducted in this dual space across all clients. FedDualAvg can be used to solve federated constrained optimization problems when regularizers are indicator functions defined over the constraints set. However, FedDualAvg uses constant learning rate as FedAvg, thus suffers the same slow convergence rate. In fact, it is an open question *if adaptive learning rates can be used to accelerate the solving of federated constrained optimizations.*

In this work, we propose the **Fed**erated **D**ual-averaging **A**daptive-gradient (**FedDA**), a general adaptive gradient framework for federated constrained optimization. FedDA is based on the dynamic mirror descent view of adaptive gradients (Huang et al., 2021). In fact, suppose we have an adaptive matrix $H$ such that it is diagonal and its $i_{th}$ diagonal element represents the adaptive gradient for the $i_{th}$ coordinate of the model parameter. Then if we view the parameter space as the primal space and the matrix $H$ as a linear mirror map, the adaptive gradient update step is equivalent to a mirror descent update step. Since adaptive gradients are updated at each iteration, the mirror map $H$ is also dynamic. Our FedDA is inspired by this mirror descent view of adaptive gradients.

In FedDA, the server maintains the global adaptive matrix and the global model weight. In each epoch, each client receives the current global adaptive matrix and global model weight. Then it performs multiple steps of 'dual-averaging' style mirror descent (Nesterov, 2009). More specifically, the client aggregates dual states, and only recover model weight (primal states) through the adaptive matrix (mirror map) when the gradient query is needed. Note that the adaptive matrix is fixed during local updates. After local updates, the server averages the dual states and then updates the global primal state and the adaptive matrix. There are two important characteristics of FedDA. Firstly, we fix the adaptive matrix during local updates to makes sure that all clients share the same dual space. Secondly, the aggregation and averaging is performed in the dual space instead of the primal space. In fact, in the constrained case, the mapping from dual space to the primal space involves the non-linear projection operation. Although averaging dual states leads to an unbiased estimate of the global gradient, averaging primal states leads to biased estimation due to projection. In summary, FedDA adopts the restarted dual-averaging strategy, where the adaptive matrix (mirror map) is refreshed at each global epoch and clients perform dual averaging locally with a fixed mirror map. Our **FedDA** is a general framework: it is flexible to the choices of adaptive gradient methods and also can combine with various gradient estimation methods. Furthermore, although it is designed for the constrained

case, FedDA works well in the unconstrained FL setting too. We highlight our **contribution** as follows:

(i) We propose **FedDA**, a fast adaptive gradient framework for constrained federated optimization problems. The framework is based on a restarted dual averaging technique and incorporates a large family of adaptive gradient methods.

(ii) **FedDA-MVR**, an instantiation of our framework, obtains the sample complexity of $\tilde{O}(K^{-1}\epsilon^{-1.5})$ and communication complexity of $\tilde{O}(K^{-0.25}\epsilon^{-1.25})$. The iteration complexity matches the optimal rate of non-adaptive federated algorithms. **FedDA-MVR** uses the momentum-based variance-reduction gradient estimation, and exponential moving average of the gradient square as adaptive learning rates.

(iii) We empirically verify the efficacy of the FedDA in solving biomarker identification tasks: the colorrectal cancer prediction task and the splice site detection task. Furthermore, we also verify the efficacy of FedDA over non-constrained FL tasks through classification tasks over CIFAR-10 and FEMNIST datasets.

**Notations.** $\nabla f(x)$ denotes the first-order derivatives of the function $f(x)$ *w.r.t.* variable $x$. $\xi$ denotes a random sample and $\nabla f(x;\xi)$ is the stochastic estimate $\nabla f(x)$. $O(\cdot)$ is the big O notation, and $\tilde{O}(\cdot)$ hides logarithmic terms. $I_d$ denotes a $d$-dimensional identity matrix. $Diag(x)$ denotes the matrix whose diagonal is the vector $x$. $\|\cdot\|$ denotes the $\ell_2$ norm for vectors and the spectral norm for matrices, respectively. $\langle\cdot,\cdot\rangle$ denotes the Euclidean inner product. [K] denotes the set of $\{1,2,...,K\}$. For a random variable $X$, $\mathbb{E}[X]$ denotes its expectation.

## 2 RELATED WORKS

**Federated Learning.** Federated Learning (McMahan et al., 2017) defines the task to learn from a set of distributed located clients under the coordination of a server. McMahan et al. (2017) proposed the FedAvg algorithm where each client performs multiple steps of gradient descent with its local data and then sends the updated model to the server for averaging. The FedAvg algorithm resembles the Local-SGD algorithm, which is studied in a more general distributed setting for a longer time (Mangasarian & Solodov, 1993). The convergence of the local-SGD method has been heavily analyzed in the literature (Stich, 2018; Karimireddy et al., 2019; Dieuleveut & Patel, 2019; Khaled et al., 2020; Yu et al., 2019; Woodworth et al., 2020; Woodworth, 2021; Glasgow et al., 2022). Various acceleration methods of FedAvg are considered and we list a few representatives here. Karimireddy et al. (2020b) adopted the idea of variance reduction technique for non-distributed finite sum problems: a 'control variate' which contains historical full gradient information is used to correct the bias of local gradients. Karimireddy et al. (2020a) proposed a general framework (MIME) to translate a centralized optimizer into the FL setting, including adaptive gradient methods. In Das et al. (2020); Khanduri et al. (2021a), momentum-based variance reduction is applied to the FL setting to control the noise of the stochastic gradients. In Das et al. (2020), a server momentum state and a client momentum state are maintained, while in Khanduri et al. (2021b), a momentum state and is matinained the momentum was averaged periodically similar to the primal state.

Adaptive gradient methods are also studied in the FL setting. The 'Adaptive Federated Optimization' (Reddi et al., 2020) method proposed to use adaptive gradients on the server side while the local gradients are used to update the states of the adaptive gradient methods. Tong et al. (2020); Wang et al. (2022) extend the method in Reddi et al. (2020) to include the AMSGrad method. In Chen et al. (2020b), the authors first showed the divergence of a naive local AMSGrad method that directly averages the primal states periodically. The authors then proposed Local-AMSGrad, a method in which clients update adaptive learning rates locally and average at the synchronization step. At the server average step, both primal states and local adaptive learning rates are averaged to replace the old states. More recently, Chen et al. (2020b); Wu et al. (2022) proposed to use fixed adaptive learning rates locally. Finally, another line of research Tang et al. (2020; 2021); Lu et al. (2022); Chen et al. (2020a) considers federated adaptive learning rates through the compression approach, these methods communicate local gradients at every step, but the compression techniques are used to reduce the communication cost. All these methods study federated adaptive methods in the unconstrained case. For solving problems with constraint in FL, a related work is (Yuan et al., 2021), where authors propose a modified local-SGD method based on the dual-averaging, however, it does not support the adaptive gradient methods.

**Adaptive Gradients in the Non-distributed Learning.** Adaptive gradient methods are widely used in the non-distributed machine learning setting. Adagrad Duchi et al. (2011) was one of the the first adaptive gradient method proposed in literature, and was shown to outperform SGD in the sparse gradient setting. Since Adagrad does not perform well under dense gradient setting and non-convex setting, some of its variants are proposed, such as SC-Adagra (Mukkamala & Hein, 2017) and SAdagrad (Chen et al., 2018b). Furthermore, Adam (Kingma & Ba, 2014) and YOGI (Zaheer et al., 2018) proposed to use the exponential moving average instead of the arithmetic average used in Adagrad. Adam/YOGI is widely used in deep learning applications; however, Adam diverges in some settings and the gradient information quickly disappears, so AMSGrad (Reddi et al., 2018) is proposed, and it applies an extra 'long term memory' variable to preserve the past gradient information to handle the convergence issue of Adam. The convergence of Adam-type methods is also studied in the literature (Chen et al., 2019; Zhou et al., 2018; Liu et al., 2019; Guo et al., 2021; Huang et al., 2021). Adaptive gradient methods with good generalization performance are also proposed, such as AdamW (Loshchilov & Hutter, 2018), Padam (Chen et al., 2018a), Adabound (Luo et al., 2019), Adabelief (Zhuang et al., 2020) and AaGrad-Norm (Ward et al., 2019).

## 3 PRELIMINARIES

In this section, we introduce some preliminaries. We consider the following formulation of Federated Learning (FL) with $K$ clients:

$$\min_{x \in \mathcal{X} \subset \mathbb{R}^d} \left\{ f(x) := \frac{1}{K} \sum_{k=1}^{K} \left\{ f^{(k)}(x) := \mathbb{E}_{\xi^{(k)} \sim \mathcal{D}^{(k)}}[f^{(k)}(x; \xi^{(k)})] \right\} \right\}. \tag{1}$$

For the $k_{th}$ client, we optimize the loss objective $f^{(k)}(x) : \mathcal{X} \to \mathbb{R}$ which is smooth and possibly non-convex, and $x$ denotes the variable of interest. $\mathcal{X} \subset \mathbb{R}^d$ is a compact and convex set. $\xi^{(k)} \sim \mathcal{D}^{(k)}$ is a random example that follows an unknown data distribution $\mathcal{D}^{(k)}$. The formulation in equation 1 includes both the homogeneous case *i.e.* $f^{(k)}(x) = f^{(j)}(x)$ for any $k, j \in [K]$, and the heterogeneous case *i.e.* $f^{(k)}(x) \neq f^{(j)}(x)$ for some $k, j \in [K]$.

Next, we introduce some basics of adaptive gradient methods from a mirror-descent perspective. Generally, mirror descent is associated with a mirror map $\Phi(x)$. Given the objective $f(x)$ and the primal state $x_t \in \mathcal{X}$ at $t_{th}$ step, we first map the primal state to the mirror space via the mirror map $y_t = \nabla\Phi(x_t)$, then we perform the gradient descent step in the mirror space: $y_{t+1} = y_t - \eta \nabla f(x_t)$, where $\eta$ is the learning rate, finally, we map $y_{t+1}$ back to the primal space as $x_{t+1} = \arg\min_{x \in \mathcal{X}} D_\Phi(x, \hat{x}_{t+1})$, where $y_{t+1} = \nabla\Phi(\hat{x}_{t+1})$ and $D_\Phi(x, y)$ denotes the Bregman Divergence associated to $\Phi$: $D_\Phi(x, y) = \Phi(x) - \Phi(y) - \langle \nabla\Phi(y), x - y \rangle$. Equivalently, the mirror descent step can also be written as a Bregman proximal gradient step as follows: $x_{t+1} = \arg\min_{x \in \mathcal{X}} \eta\langle \nabla f(x_t), x \rangle + D_\Phi(x, x_t)$. For the adaptive gradient methods, we uses the following mirror map: $\Phi(x) = \frac{1}{2}x^T H x$, where $H$ is a positive definite adaptive matrix. Many adaptive gradient methods can be written in the following proximal gradient descent form:

$$x_{t+1} = \arg\min_{x \in \mathcal{X}} \eta\langle \nu_t, x \rangle + \frac{1}{2}(x - x_t)^T H_t (x - x_t), \tag{2}$$

we replace the gradient $\nabla f(x)$ with the generalized gradient estimation $\nu_t$, besides, we replace $H$ with $H_t$ based on the fact that the adaptive matrix is updated at every step. Next, we show some examples of adaptive gradients methods that can be phrased as the above formulation. For the Adagrad (Duchi et al., 2011) method, we set

$$\nu_t = \nabla f(x_t, \xi_t), \; H_t = Diag(\sqrt{\mu_t}), \; \mu_t = \frac{1}{t}\sum_{i=1}^{t} \nu_i^2 \tag{3}$$

For Adam (Kingma & Ba, 2014), we have:

$$\hat{\nu}_t = (1 - \beta_1)\nabla f(x_t, \xi_t) + \beta_1 \hat{\nu}_{t-1}, \hat{\mu}_t = (1 - \beta_2)\nabla f(x_t, \xi_t)^2 + \beta_2 \hat{\mu}_{t-1}$$
$$\nu_t = \hat{\nu}_t/(1 - \gamma_1^t), \; \mu_t = \hat{\mu}_t/(1 - \gamma_2^t), H_t = Diag(\sqrt{\mu_t} + \epsilon) \tag{4}$$

where $\beta_1, \beta_2, \gamma_1, \gamma_2$ are some constants.

---

**Algorithm 1 FedDA-Server**

1: **Input:** Number of global epochs $E$, size of the first mini-batch $b_1$, tuning parameters $\{\beta_\tau\}_{i=1}^{E}$;
2: **Initialize:** Choose $x_0 \in \mathcal{X}$ and each client selects a mini-batch samples $\mathcal{B}_0^{(k)}$ of size $b_1$ to evaluate $\nabla f^{(j)}(x_0, \mathcal{B}_0^{(k)})$ locally and the server averages local gradients to compute $\nu_0$;
3: **for** $\tau = 0$ **to** $E - 1$ **do**
4:    **for** the client $k \in [K]$ in parallel **do**
5:       $(z_{\tau+1,I}^{(k)}, \nu_{\tau+1,I}^{(k)})$ = **FedDA**-client$(x_\tau, \nu_\tau, H_\tau)$
6:    **end for**
7:    Compute $z_{\tau+1} = \frac{1}{K} \sum_{k=1}^{K} z_{\tau+1,I}^{(k)}$;
8:    Compute $x_{\tau+1} = \underset{x \in \mathcal{X}}{\arg\min}\{-\langle x, z_{\tau+1}\rangle + \frac{1}{2}(x - x_\tau)^T H_\tau(x - x_\tau)\}$;
9:    Compute $\nu_{\tau+1} = \frac{1}{K} \sum_{k=1}^{K} \nu_{\tau+1,I}^{(k)}$, $H_{\tau+1} = \mathcal{V}(H_\tau, z_{\tau+1}, \beta_\tau)$;
10: **end for**

---

**Algorithm 2 FedDA-Client $(x_\tau, \nu_\tau, H_\tau)$**

1: **Input:** Number of local steps $I$, mini-batch size $b$, tuning parameters $\{\eta_{\tau+1,i}\}_{i=0}^{I-1}$, $\{\alpha_{\tau+1,i}\}_{i=1}^{I}$;

2: **Initialize:** $x_{\tau+1,0}^{(k)} = x_\tau$; $\nu_{\tau+1,0}^{(k)} = \nu_\tau$; $z_{\tau+1,0}^{(k)} = 0$;
3: **for** $i = 0$ **to** $I - 1$ **do**
4:    Compute $z_{\tau+1,i+1}^{(k)} = z_{\tau+1,i}^{(k)} - \eta_{\tau+1,i}\nu_{\tau+1,i}^{(k)}$;
5:    Compute $x_{\tau+1,i+1}^{(k)} = \underset{x \in \mathcal{X}}{\arg\min}\{-\langle x, z_{\tau+1,i+1}^{(k)}\rangle + \frac{1}{2}(x - x_{\tau+1,0}^{(k)})^T H_\tau(x - x_{\tau+1,0}^{(k)})\}$;
6:    Compute $\nu_{\tau+1,i+1}^{(k)} = \mathcal{U}(\nu_{\tau+1,i}^{(k)}, x_{\tau+1,i+1}^{(k)}, x_{\tau+1,i}^{(k)}; \alpha_{\tau+1,i+1}, \mathcal{B}_{\tau+1,i+1}^{(k)})$, where $\mathcal{B}_{\tau+1,i+1}^{(k)}$ is a minibatch of size $b$ of random samples from the client $k$;
7: **end for**
8: **Output:** Send $z_{\tau+1,I}^{(k)}$, $\nu_{\tau+1,I}^{(k)}$ to the server.

---

## 4   Local Adaptive Gradients via Dual Averaging

In this section, we introduce **FedDA**, a fast adaptive gradient framework for federated constrainted optimization problems. The procedure of **FedDA** is summarized in Algorithm 1. In Algorithm 1, we perform $E$ global steps and at each global step, every client runs Algorithm 2.

In Algorithm 2, clients receive the current model weight $x_\tau$, gradient estimation $\nu_\tau$ and adaptive gradient matrix $H_\tau$. The clients then perform $I$ local training steps: line 3- line 7 in Algorithm 2. For each step, we first accumulate the dual state in the variable $z_{\tau,i}^{(k)}$ (line 4), then we calculate the local primal state $x_{\tau,i}^{(k)}$ (line 5), which is a proximal gradient step similar to equation 2. The function of this step is to map the aggregated dual state $z_{\tau,i}^{(k)}$ back to the primal space, and we use the primal state to query the gradient to update the estimation of the gradient $\nu_{\tau,i}^{(k)}$ (line 6). Note that we use a fixed adaptive matrix $H_\tau$ during local steps, this makes the clients share the same dual space. In line 6 of Algorithm 2, we update the gradient estimation $\nu_{\tau,i}^{(k)}$. The update rule $\mathcal{U}(\cdot)$ is general, *e.g.*,the momentum-based variance reduction update equation 5 and the momentum update equation 6 as follows ($\alpha_{\tau,i}$ is a momentum coefficient):

$$\nu_{\tau+1,i+1}^{(k)} = \nabla f^{(k)}(x_{\tau+1,i+1}^{(k)}, \mathcal{B}_{\tau+1,i+1}^{(k)}) + (1 - \alpha_{\tau+1,i+1})(\nu_{\tau+1,i}^{(k)} - \nabla f^{(k)}(x_{\tau+1,i}^{(k)}, \mathcal{B}_{\tau+1,i+1}^{(k)})) \tag{5}$$

and

$$\nu_{\tau+1,i+1}^{(k)} = \alpha_{\tau+1,i+1}\nabla f^{(k)}(x_{\tau+1,i+1}^{(k)}, \mathcal{B}_{\tau+1,i+1}^{(k)}) + (1 - \alpha_{\tau+1,i+1})\nu_{\tau+1,i}^{(k)} \tag{6}$$

After the client finishes Algorithm 2, it returns the aggregated local dual states $z_{\tau+1,I}^{(k)}$ and the local gradient estimation $\nu_{\tau+1,I}^{(k)}$ to the server. The server first averages the local dual states (line 8 of

Algorithm 1) to get $z_{\tau+1}$. We can average local dual states as all clients have a common dual space. The server then calculates the new primal states $x_{\tau+1}$ as in line 9 of Algorithm 1. Next, the gradient estimation $\nu_\tau$ is also updated by averaging local gradient estimates (line 9 of Algorithm 1). Finally, we update the adaptive matrix $H_\tau$ (line 9 of Algorithm 1). The update rule $\mathcal{V}$ is general, *e.g.*,

$$\mu_{\tau+1} = \beta_{\tau+1} z_{\tau+1}^2 / \eta_{\tau+1,I-1}^2 + (1 - \beta_{\tau+1})\mu_\tau, H_{\tau+1} = Diag(\sqrt{\mu_{\tau+1}} + \epsilon) \tag{7}$$

and

$$\mu_{\tau+1} = \beta_{\tau+1} ||z_{\tau+1}|| / \eta_{\tau+1,I-1} + (1 - \beta_{\tau+1})\mu_\tau, H_{\tau+1} = (\mu_{\tau+1} + \epsilon)I_d \tag{8}$$

where we set $\mu_0 = 0$, $\epsilon$ is some constant. In summary, Algorithm 1 aggregates and averages dual states at each global round. The adaptive matrix $H_\tau$ is fixed during local updates and is refreshed on the server side at each global round. Since the algorithm uses a new mirror map (adaptive gradient matrix) at each global round, we name the strategy as restarted dual averaging.

*Remark* 4.1. A notable characteristic of FedDA is the dual-averaging strategy in local updates, this contrasts with the 'primal averaging' strategy used by many existing adaptive FL algorithms (Karimireddy et al., 2020a). In fact, dual-averaging is essential for FedDA to solve federated constrained optimization problems, due to the fact that primal and dual states are connected through a non-linear mapping (*i.e.* the projection operation).

*Remark* 4.2. By choosing different update rules $\mathcal{U}$ and $\mathcal{V}$, we can create many variants of FedDA. An representative is **FedDA-MVR**, in which we update $\nu_{\tau,i}^{(k)}$ with momentum-based variance reduction ( equation 5) and the adaptive matrix $H_\tau$ with an exponential average of the gradient square ( equation 7).

*Remark* 4.3. The argmin operation in line 8 of Algorithm 1 (line 5 of Algorithm 2) does not lead to extra computational cost. We solve it through two steps: solving the quadratic optimization problem (a generalized adaptive gradient step) and projection. The projection is necessary to make the solution feasible and for many constraints, the proximal operators are well defined.

## 5 THEORETICAL ANALYSIS

In this section, we provide the theoretical analysis of our **FedDA**. More specifically, we focus on the analysis of **FedDA-MVR**. We first state the assumptions we need in our analysis:

### 5.1 SOME MILD ASSUMPTIONS

**Assumption 5.1** (Bounded Client Heterogeneity). *The difference of gradients between different workers are bounded:* $||\nabla f^{(k)}(x) - \nabla f^{(\ell)}(x)||^2 \leq \zeta^2$, $\forall k, \ell \in [K]$.

We measure the heterogeneity of the clients in terms of gradient dissimilarity. The above assumption or its similar form is also exploited in the analysis of other FL Algorithms, such as in Khanduri et al. (2021a); Das et al. (2020).

**Assumption 5.2.** *The function $f(x)$ is bounded from below in $\mathcal{X}$,* i.e., $f^* = \inf_{x \in \mathcal{X}} f(x)$.

**Assumption 5.3** (Unbiased and Bounded-variance Stochastic Gradient). *The stochastic gradients are unbiased with bounded variance,* i.e. $\mathbb{E}[\nabla f^{(k)}(x; \xi^{(k)})] = \nabla f^{(k)}(x)$ *and there exists a constant $\sigma$ such that* $\mathbb{E}||\nabla f^{(k)}(x; \xi^{(k)}) - \nabla f^{(k)}(x)||^2 \leq \sigma^2, \forall \xi^{(k)} \sim \mathcal{D}^{(k)}, \forall k \in [K]$

Assumption 5.2 guarantees the feasibility of the Federated Learning problem equation 1, and Assumption 5.3 is widely used in stochastic optimization analysis.

**Assumption 5.4.** *The adaptive matrix $H_\tau$ is symmetric positive definite,* i.e. *there exists a constant $\rho > 0$ such that* $H_\tau \succeq \rho I_d \succ 0$, $\forall t \geq 1$,

In our analysis, we assume the adaptive matrix is positive definite, and this requirement can be easily satisfied by many adaptive gradient methods. Firstly, most adaptive gradient methods always have non-negative adaptive learning rates, such as equation 3 and equation 4. To make it positive, we can add a bias term $\epsilon$ such as in the Adam update rule equation 4.

**Assumption 5.5** (Sample Gradient Lipschitz Smoothness). *The stochastic functions $f^{(k)}(x, \xi^{(k)})$ with $\xi^{(k)} \sim \mathcal{D}^{(k)}$ for all $k \in [K]$, satisfy the mean squared smoothness property, i.e, we have* $\mathbb{E}||\nabla f^{(k)}(x; \xi^{(k)}) - \nabla f^{(k)}(y; \xi^{(k)})||^2 \leq L^2 ||x - y||^2$ *for all $x, y \in \mathbb{R}^d$*

The smoothness assumption above is a slightly stronger requirement than the standard smooth condition, but this assumption is widely used in the analysis of variance reduction methods, such as SPIDER Fang et al. (2018) and STORM Cutkosky & Orabona (2019).

## 5.2 Convergence Property of **FedDA-MVR**

In this subsection, we study the convergence property of our **FedDA-MVR** variant. For convenience of discussion, we **redefine the subscript** $t = \tau I + i$, *i.e.* we denote the $t$ step as the $i$ local step in the $\tau$ global round. Similarly, we denote the total number of running steps as $T = EI$. We analyze our algorithm through the following measure:

$$\mathcal{G}_t = \frac{\rho^2}{\eta_t^2}||\tilde{x}_t - \tilde{x}_{t+1}||^2 + ||\bar{\nu}_t - \nabla f(\tilde{x}_t)||^2 \tag{9}$$

where $\bar{\nu}_t$ denotes the average gradient estimation at the $t_{th}$ step and $\tilde{x}_t$ denotes the virtual global primal state at the $t_{th}$ step (see Section B.2 in the appendix for formal definitions). In Remark B.18 of the appendix, we discuss the intuition of the measure $\mathcal{G}_t$. In particular, in the unconstrained case *i.e.* when $\mathcal{X} = R^d$, the measure upper-bounds the square norm of the gradient. Therefore, the convergence of our measure $\mathcal{G}_t$ means the convergence to a first-order stationary point. Now, we are ready to provide the main result of our convergence theorem.

**Theorem 5.6.** *In Algorithm 1, given the parameter* $\kappa = \frac{\rho K^{2/3}}{L}$, $c = \frac{96L^2}{K\rho^2} + \frac{\rho}{72\kappa^3 LK^{0.5}I^2}$, $w = \max\{2, 48^3 I^6 K^{3.5}\}$, $b_1 \geq 1$, $b \geq 1$, $\beta > 0$ *and choose learning rate* $\eta_t = \frac{\kappa}{(w_t + t + I)^{1/3}}$, *the momentum coefficient* $\alpha_t = c\eta_t^2$, *the adaptive gradient coefficient* $\beta_t = \beta$ *and the mini-batch size of* $b_1$ *for the first iteration and* $b$ *for other iterations, then we have:*

$$\frac{1}{T}\sum_{t=0}^{T-1}\mathbb{E}[\mathcal{G}_t] \leq \left[\frac{96LK^{0.5}I^2}{T} + \frac{2L}{K^{2/3}T^{2/3}}\right](f(x_0) - f^*) + \left[\frac{72KI^4}{b_1 T} + \frac{3K^{0.5}I^2}{2b_1 K^{2/3}T^{2/3}}\right]\sigma^2$$

$$+ 192^2 \times \left(\frac{48K^{0.5}I^2}{T} + \frac{1}{K^{2/3}T^{2/3}}\right) \times \left(\frac{\sigma^2}{4b} + \frac{2\zeta^2}{21}\right)\log(T+1).$$

Note, by choosing a proper value of local updates $I$ and using a minibatch of samples for the first iteration to decrease the noise, our result matches the best known convergence rate for stochastic federated gradient methods (Khanduri et al., 2021a), *i.e.* our algorithms has sample complexity of $\tilde{O}(\epsilon^{-1.5})$ with a linear speed up *w.r.t* the number of clients $K$. More formally, we have:

**Corollary 5.7.** *Suppose in Algorithm 1, we set* $I = O((T/K^{3.5})^{1/6})$, *and use sample minibatch of size* $O(K^{0.5}I^2)$ *in the initialization, then we have:* $\frac{1}{T}\sum_{t=1}^{T}\left(\mathbb{E}[\mathcal{G}_t]\right) = \tilde{O}(K^{-2/3}T^{-2/3})$, *and to reach an* $\epsilon$-*stationary point, we need to make* $\tilde{O}(K^{-1}\epsilon^{-1.5})$ *number of steps and need* $\tilde{O}(K^{-0.25}\epsilon^{-1.25})$ *number of communication rounds.*

As shown by Corollary 5.7, FedDA-MVR achieves iteration complexity $O(K^{-1}\epsilon^{-1.5})$ which matches the optimal rates of non-convex stochastic optimization (Arjevani et al., 2019). In contrast, FedDualAvg (Yuan et al., 2021) shows the convergence rate of $O(\epsilon^{-2})$ for the general non-convex objective under the strong bounded gradient assumption. Our FedDA-MVR achieves a faster convergence rate and does not require the bounded gradient assumption. As for the communication complexity, we reach $\tilde{O}(K^{-0.25}\epsilon^{-1.25})$. Some methods achieve $O(\epsilon^{-1})$ communication rate such as STEM (Khanduri et al., 2021a) and FAFED (Wu et al., 2022). However, STEM does not consider the adaptive gradient methods nor the constrained setting; FAFED considers the adaptive gradient methods, but it requires the strong assumption of bounded gradient and only considers the unconstrained setting.

## 6 Experiments

In this section, we perform experiments to verify the efficacy of our proposed **FedDA** on federated biomarker identification task and the general classification tasks. More specifically, we consider the variant of **FedDA-MVR** here, and defer experiments for other variants to Section A of the appendix.

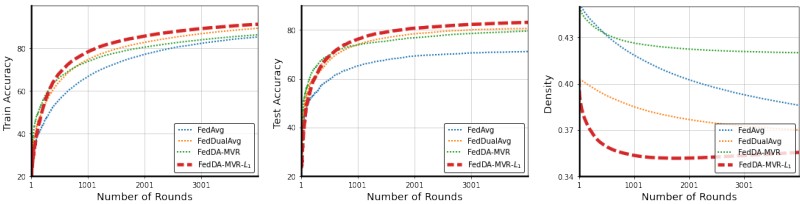

Figure 1: Results for the PATHMNIST Yang et al. (2021) dataset. Plots show the Train Accuracy, Test Accuracy, Density of Model Parameters vs Number of Rounds ($E$ in Algorithm 1) respectively. The post-fix of $L_1$ means the $L_1$ constraints. Number of local steps $I$ is chosen as 5.

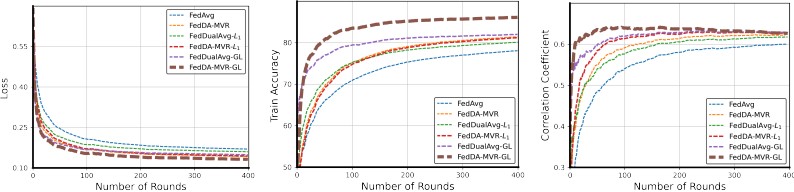

Figure 2: Results for the MEMset Donar Dataset Meier et al. (2008). Plots show the Train Loss, Train Accuracy and the Pearson Correlation Coefficient (between prediction and targets). The post-fix of $L_1$ means $L_1$ constraints and $GL$ means Group Lasso constraints. Number of local steps $I$ is 5.

We consider three sets of experiments: Colorrectal Cancer Survival Prediction, Splice Site Detection and a general multiclass image classification task. All experiments are run on a machine with an Intel Xeon Gold 6248 CPU and 4 Nvidia Tesla V100 GPUs. The code is written in Pytorch. We simulate the Federated Learning environment through the Pytorch.distributed package.

## 6.1 COLORRECTAL CANCER SURVIVAL PREDICTION WITH BIOMARKER IDENTIFICATION

In this subsection, we consider a colorrectal cancer prediction task on the PATHMNIST dataset (Yang et al., 2021; Kather et al., 2019), which contains 9 different classes and 89996 training images. We equally randomly split the training set into 10 clients. We use the original test set for the metric. In this task, we impose the $L_1$ sparsity constraint to identify biomarkers. We compare with the following baselines: FedAvg (McMahan et al., 2017) and FedDualAvg (Yuan et al., 2021). For our FedDA-MVR, we train with and without the $L_1$ constraint.

The results are summarized in Figure 1, the plots are averaged over 5 independent runs and then smoothed. In Figure 1, FedDualAvg and FedDA-MVR-$L_1$ consider the $L_1$ constraint, while FedAvg and FedDA-MVR do not. We show results of Train/Test Accuracy and also the number of non-zero (below a threshold) elements in the parameter (the rightmost plot in Figure 1). As shown in the plots, FedDA-MVR-$L_1$ outperforms unconstrained FedDA-MVR in all metrics, in particular, FedDA learns a much sparser model and therefore can better identify important factors for cancer survival than other methods. Furthermore, FedDA-MVR-$L_1$ also outperforms FedAvg and FedDualAvg in all metrics. This shows that our algorithm can effectively exploit adaptive gradient information in the constrained case. For more details of this experiment, please refer to Section A.1 of the appendix.

## 6.2 SPLICE SITE DETECTION WITH BIOMARKER IDENTIFICATION

In this subsection, we consider a splice site detection task on the MEMset Donar Dataset (Meier et al., 2008). Splice sites are the regions between coding (exons) and non-coding (introns) DNA segments. Splice site detection plays an important role in gene finding. We follow the train/test split in (Meier et al., 2008) and then randomly split the training set to 10 clients. Group Lasso is widely used to solve the splice site detection problem, so we use FedDA-MVR with the Group Lasso constraint in the experiments. For the baselines, we compare with FedAvg (McMahan et al., 2017), FedDualAvg Yuan et al. (2021) and FedDA-MVR without constraints. For FedDualAvg, we consider the $L_1$ constraint and Group Lasso constraint.

The results are summarized in Figure 2, the plots are averaged over 5 independent runs and then smoothed. As shown in the plots, FedDA-MVR-$GL$ outperforms unconstrained FedDA-MVR and

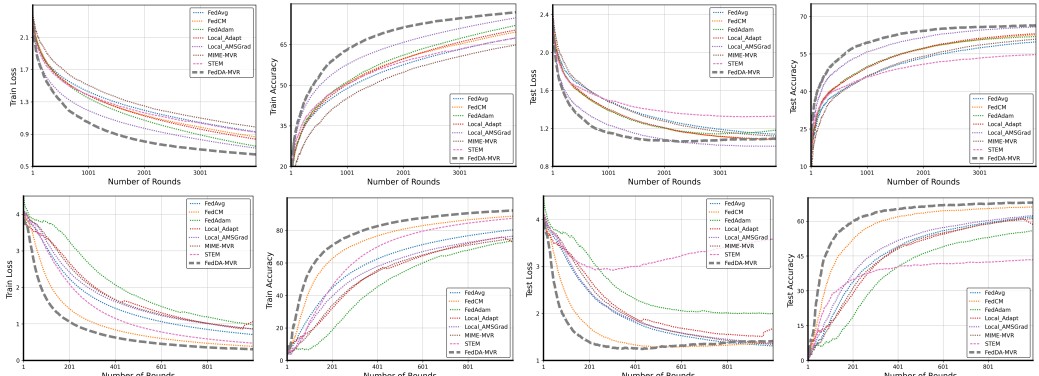

Figure 3: Results for (Homogeneous) CIFAR10 dataset (Top) and FEMNIST (Bottom). From left to right, we show Train Loss, Train Accuracy, Test Loss, Test Accuracy *w.r.t* the number of rounds (E in Algorithm 1), respectively. $I$ is chosen as 5.

FedDA-MVR-$L_1$. In fact, FedDA-MVR-$GL$ gets better performance by identifying meaningful feature groups. Furthermore, FedDA-MVR-$GL$ also outperforms FedAvg and FedDualAvg-$GL$ (FedDualAvg-$L_1$). This shows that our algorithm can effectively exploit acceleration of adaptive gradients. For more details of this experiment, please refer to Section A.1 of the appendix.

## 6.3 IMAGE CLASSIFICATION TASK WITH CIFAR10 AND FEMNIST

FedDA is a general framework for FL and can also solve unconstrained tasks. In this subsection, we consider an unconstrained image classification task. More specifically, we consider two datasets: CIFAR10 (Krizhevsky et al., 2009) and FEMNIST (Caldas et al., 2018). We construct both homogeneous and heterogeneous cases based on CIFAR10. For the homogeneous case, we uniformly randomly distribute them into 10 clients and for the heterogeneous case, we create imbalanced class distribution among clients (please see Appendix A.3). FEMNIST is a Federated dataset of hand-written digits; it contains hand-written digits of 3550 users. Data distribution of FEMNIST is heterogeneous for different writing styles of people. We compare our method with following baselines: non-adaptive methods: FedAvg (McMahan et al., 2017), FedCM (Xu et al., 2021b), STEM (Khanduri et al., 2021a) and adaptive methods: FedAdam (Reddi et al., 2020), Local-Adapt (Wang et al., 2021), Local-AMSGrad (Chen et al., 2020b), MIME-MVR (Karimireddy et al., 2020a).

For all methods, we tune their hyper-parameters to find the best setting. The results are summarized in Figure 3, the plots are averaged over 5 runs and then smoothed. As shown in the figures, our FedDA-MVR outperforms all baselines. We observe that adaptive methods in general get better train and test performance. Finally, the superior performance of our method compared with the three adaptive baselines shows that our method exploits adaptive information better; for example, MIME-MVR also exploits the momentum-based variance reduction technique, but it fixes all optimizer states during local updates, in contrast, we only fix the adaptive matrix but update the momentum $\nu_t^{(k)}, k \in [K]$ at every step. For the full set of experiments, please refer to Section A.1 and A.2 of the appendix.

## 7 CONCLUSION

In this paper, we proposed FedDA, an adaptive gradient framework for federated constrained optimization. FedDA incorporates various adaptive gradients and momentum-based acceleration methods. More specifically, we adopt the Mirror Descent view of adaptive gradients and propose to maintain and average the dual states in the training, meanwhile we fix the adaptive matrix during local training such that the dual spaces are aligned among clients. We also analyze the convergence property of our Framework: for the variant FedDA-MVR, we proved that it reaches an $\epsilon$-optimal stationary point with $\tilde{O}(K^{-1}\epsilon^{-1.5})$ gradient queries and $\tilde{O}(K^{-0.25}\epsilon^{-1.25})$ communication rounds. Finally, we validate our algorithm for both biomarker identification tasks and general unconstrained image classification tasks. The numerical results show the superior performance of our algorithm compared to various baseline methods.

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
