(Praneeth Karimireddy et al., 2020). FedAMS (Wang et al., 2022) and FedAMSGrad (Tong et al., 2020) get similar performance as FedAdam, so we omit it in the plots.

For all methods, we tune their hyper-parameters to find the best setting. The results are summarized in Figure 3, the plots are averaged over 5 runs and then smoothed. As shown in the figures, our FedDA-MVR outperforms all baselines. In addition, the FedAvg algorithm has competitive training performance; however, it tends to overfit the training data severely and suffers most from the heterogeneity. Then we observe that adaptive methods in general get better train and test performance. Finally, the superior performance of our method compared with the three adaptive baselines shows that our method exploits adaptive information better; for example, MIME-MVR also exploits the momentum-based variance reduction technique, but it fixes all optimizer states during local updates, in contrast, we only fix the adaptive matrix but update the momentum $\nu_t^{(k)}, k \in [K]$ at every step. For the full set of experiments, please refer to Section A.1 and A.2 of the appendix.

## 7 CONCLUSION

In this paper, we proposed the FedDA an adaptive gradient framework for federated constrained optimization. FedDA incorporates various adaptive gradients and momentum-based acceleration methods. More specifically, we adopt the Mirror Descent view of adaptive gradients and propose to maintain and average the dual states in the training, meanwhile we fix the adaptive matrix during local training such that the dual spaces are aligned among clients. We also analyze the convergence property of our Framework: for the variant FedDA-MVR, we proved that it reaches an $\epsilon$-optimal stationary point with $\tilde{O}(K^{-1}\epsilon^{-1.5})$ gradient queries and $\tilde{O}(K^{-0.25}\epsilon^{-1.25})$ communication rounds. Finally, we validate our algorithm for both biomarker identification tasks and general unconstrained image classification tasks. The numerical results show the superior performance of our algorithm compared to various baseline methods.

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

## A EXPERIMENTAL DETAILS AND RESULTS

In this section, we add additional experiments. In Section A.1, we consider more variants of **FedDA** besides **FedDA-MVR**. More specifically, we consider four variants of **FedDA**. We introduce two cases for the update of the adaptive matrix $H_\tau$ in equation 7 and equation 8 and we denote them as case 1 and case 2, similarly, we denote equation 5 and equation 6 as case 1 and case 2 of gradient estimation respectively. So we have four different variants, we denote them as **FedDA-$i$-$j$**, for $i, j \in \{1, 2\}$, where $i$ shows the choice of gradient estimation and $j$ shows the choice of adaptive matrix update rule. Note **FedDA-MVR** corresponds to **FedDA-1-1** as we choose Case 1 of gradient estimation and Case 1 of adaptive matrix update in Algorithm 1. We also introduce more details such as the hyper-parameter choices. Then in Section A.2, we perform some ablation studies and compare our FedDA with other baselines in more detail; In Section A.3, we include experiments when we construct heterogeneous dataset from CIFAR10; Finally in Section A.4, we show the form of our FedDA when $I = 1$, i.e. no local steps.

### A.1 EXPERIMENTAL RESULTS FOR MORE VARIANTS OF FEDDA

#### A.1.1 COLORRECTAL CANCER SURVIVAL PREDICTION WITH BIOMARKER IDENTIFICATION

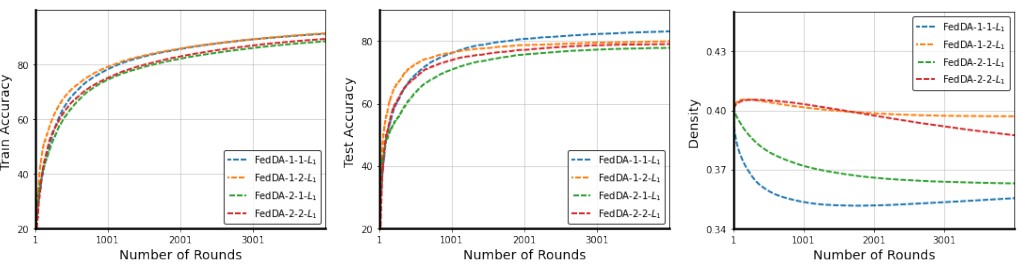

Figure 4: Results for the PATHMNIST dataset. Plots show the Train Accuracy, Test Accuracy, Density vs Number of Rounds ($E$ in Algorithm 1) respectively. The post-fix of $L_1$ means we consider the $L_1$ constraints.

In this task, for $L_1$ constraint, we consider the constraint set of $|x|_1 \le \epsilon$, where $x$ is the model parameter and $\epsilon$ is some constant. We use a 4-layer convolutional neural network with 32 filters at each layer. We have 10 clients and run 20000 steps ($T$), average states with interval 5 ($I$) and use mini-batch size of 16. Besides, we calculate density with threshold 0.01. For other hyper-parameters, we perform grid search and choose the best setting for each method. More specifically, for the SGD method, we use learning rate 0.01; for the FedDualAvg algorithm, we use local learning rate 0.1, global learning rate 0.1, $L_1$ constraint 0.01; for our FedDA-MVR, we use learning rate 0.01, $w$ as 100000, $c$ as 5000000, $\beta$ as 0.999 and $\tau$ as 0.01, for the $L_1$ regularized version FedDA-MVR-$L_1$, we also add $L_1$ constraint $\epsilon = 0.01$. For other variants of FedDA: for FedDA-2-1, we use learning rate 0.001, $\alpha$ as 0.9, $\beta$ as 0.999, $\tau$ as 0.01; for FedDA-1-2, we use learning rate 1, $w$ as 10000, $c$ as 200, $\beta$ as 0.999, $\tau$ as 0.001, $L_1$ constraint 0.01; for FedDA-2-2, we use learning rate 0.01, $\alpha$ 0.9, $\beta$ as 0.999, $\tau$ as 0.01, $L_1$ constraint 0.01.

The experimental results for different variants of FedDA is summarized in Figure 4. As shown by the plots, all variants of FedDA get good performance, but we find FedDA-MVR (FedDA-1-1) gets most sparse model as measured by the density metric.

#### A.1.2 SPLICE SITE DETECTION WITH BIOMARKER IDENTIFICATION

In this task, for group lasso constraint, we consider the constraint set of $\sum_{i=1}^{q} |x_q|_2 \le \epsilon$, where $x$ is the model parameter, $q$ is the number of groups, $x_q$ denotes a subset of parameters of group $q$, $\epsilon$ is some constant. The MEMset donor data set consists of a training set of 8415 true and 179,438 false human donor sites, and a test set of 4208 true and 89,717 false donor sites. A sequence of a real splice site consists of the last three bases of the exon and the first six bases of the intron. A training sample is a sequence of length of 7 with values in $\{A, C, G, T\}$. The data are available

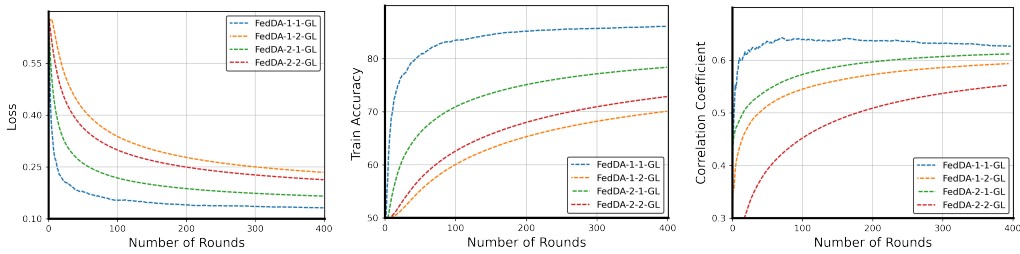

Figure 5: Results for the MEMset Donar Dataset. Plots show the Train Loss, Train Accuracy and the correlation coefficient, respectively.

at `http://hollywood.mit.edu/burgelab/maxent/ssdata/`. We follow Meier et al. (2008) to create a balanced training set with 5610 true/false samples and an unbalanced validation set with 2805 true/59804 false samples. We have 10 clients and randomly evenly distribute the training data over 10 clients. We consider the logistic regression model with group lasso constraint and include all the three-way and lower order interactions. In the experiments, we run 2000 steps ($T$), average states with interval 5 ($I$) and use mini-batch size of 16. For other hyper-parameters, we perform grid search and choose the best setting for each method. More specifically, for the FedAvg method, we use learning rate 0.1; for the FedDualAvg algorithm with $L_1$ constraint, we use local learning rate 0.1, global learning rate 0.2, $L_1$ constraint 0.01; for the FedDualAvg algorithm with group lasso constraint, we use local learning rate 0.1, global learning rate 0.5, $L_1$ constraint 0.01; for our FedDA-MVR, we use learning rate 0.1, $w$ as 10000, $c$ as 40000, $\beta$ as 0.999 and $\tau$ as 0.01, for the $L_1$ constrained version FedDA-MVR-$L_1$, we also add $L_1$ constraint 0.01, for the group lasso version FedDA-MVR-$GL$, we also add group lasso constraint of coefficient 0.01. For other variants of FedDA: for FedDA-1-2, we use learning rate 1, $w$ as 10000, $c$ as 200, $\beta$ as 0.999, $\tau$ as 0.001, group lasso constraint 0.01; for FedDA-2-1, we use learning rate 0.001, $\alpha$ as 0.9, $\beta$ as 0.999, $\tau$ as 0.01, group lasso constraint 0.01; for FedDA-2-2, we use learning rate 0.01, $\alpha$ 0.9, $\beta$ as 0.999, $\tau$ as 0.01, group lasso constraint 0.01.

The experimental results for different variants of FedDA is summarized in Figure 5. As shown by the plots, all variants of FedDA get good performance, but we find FedDA-MVR (FedDA-1-1) gets the highest test correlation coefficient among all variants. Note that the correlation coefficient is the maximum (among all possible threshold, and we find 0.95 is the best value for all methods) Pearson correlation between the binary random variable of the true class membership and the binary random variable of the predicted class membership.

### A.1.3 IMAGE CLASSIFICATION TASK WITH CIFAR10 AND FEMNIST

In this unconstrained federated image classification task, we use a 4-layer convolutional neural network with 64 filters at each layer. For the FEMNIST dataset, we randomly sample 50 users at each global round. We run 20000 steps ($T$), average states with interval 5 ($I$) and use mini-batch size of 16. For other hyper-parameters, we perform grid search and choose the best setting for each method. In the CIFAR10 related experiments, for the SGD method, we use learning rate 0.005; for the FedCM algorithm, we use learning rate 0.01, momentum coefficient $\alpha$ as 0.9; for the FedAdam algorithm, we use local learning rate 0.001, global learning rate 0.002, momentum coefficient 0.9, coefficient for adaptive matrix $\beta$ as 0.999; for the Local-Adapt algorithm, we use local learning rate 0.001, global learning rate 0.002, momentum coefficient 0.9, coefficient for adaptive matrix $\beta$ as 0.999; for the Local-AMSGrad algorithm, we use learning rate 0.001, momentum coefficient 0.9, adaptive matrix coefficient 0.999; for the MIME-MVR algorithm, we use learning rate 0.1, $w$ 100, $c$ as 2000; for the STEM algorithm, we use learning rate 0.1, $w$ 100 and $c$ 2000; for our FedDA-MVR, we use learning rate 0.02, $w$ as 10000, $c$ as 1000000, $\beta$ as 0.999 and $\tau$ as 0.01. For other variants of FedDA: for FedDA-2-1, we use learning rate 0.001, $\alpha$ as 0.9, $\beta$ as 0.999, $\tau$ as 0.01; for FedDA-1-2, we use learning rate 1, $w$ as 5000, $c$ as 100, $\beta$ as 0.999, $\tau$ as 0.01; for FedDA-2-2, we use learning rate 0.01, $\alpha$ 0.9, $\beta$ as 0.999, $\tau$ as 0.01.

Then in the FEMNIST experiments, for the SGD method, we use learning rate 0.1; for the FedCM algorithm, we use learning rate 0.1, momentum coefficient $\alpha$ as 0.9; for the FedAdam algorithm,

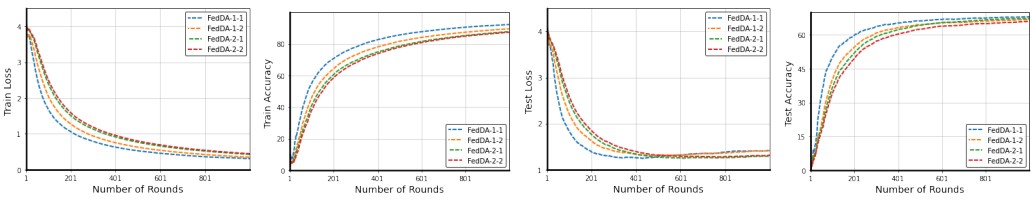

Figure 6: Results for CIFAR10 dataset. From left to right, we show Train Loss, Train Accuracy, Test Loss, Test Accuracy *w.r.t* the number of global rounds (E in Algorithm 1), respectively.

Figure 7: Results for FEMNIST dataset. From left to right, we show Train Loss, Train Accuracy, Test Loss, Test Accuracy *w.r.t* the number of global rounds (E in Algorithm 1), respectively.

we use local learning rate 0.02, global learning rate 0.04, momentum coefficient 0.9, coefficient for adaptive matrix $\beta$ as 0.999; for the Local-Adapt algorithm, we use local learning rate 0.02, global learning rate 0.02, momentum coefficient 0.9, coefficient for adaptive matrix $\beta$ as 0.999; for the Local-AMSGrad algorithm, we use learning rate 0.0005, momentum coefficient 0.9, adaptive matrix coefficient 0.999; for the MIME-MVR algorithm, we use learning rate 1, $w$ 10000, $c$ as 400; for the STEM algorithm, we use learning rate 1, $w$ 10000 and $c$ 400; for our FedDA-MVR, we use learning rate 0.02, $w$ as 10000, $c$ as 1000000, $\beta$ as 0.999 and $\tau$ as 0.01. For other variants of FedDA: for FedDA-2-1, we use learning rate 0.001, $\alpha$ as 0.9, $\beta$ as 0.999, $\tau$ as 0.01; for FedDA-1-2, we use learning rate 1, $w$ as 5000, $c$ as 100, $\beta$ as 0.999, $\tau$ as 0.01; for FedDA-2-2, we use the learning rate 0.01, $\alpha$ 0.9, $\beta$ as 0.999, $\tau$ as 0.01.

The experimental results for different variants of FedDA is summarized in Figure 6 and 7. As shown by plots, all variants of FedDA get good performance. FedDA-MVR (FedDA-1-1) gets the best performance in most metrics, we observe that its test loss show some extent of overfitting in the late training stage.

## A.2 MORE DISCUSSION OF EXPERIMENTAL RESULTS

In this subsection, we make more detailed comparison between our FedDA and other baselines (The experiments are over homogeneous CIFAR10 dataset). In Figure 8, we compare FedCM with FedDA-2-1 and FedDA-2-2 for different values of local steps $I$. Since FedDA-2-1 and FedDA-2-2 do not use variance reduction acceleration, the superior performance shows the effectiveness of using adaptive gradients in our framework. Next, In Figure 9, we compare Local-AMSGrad vs FedDA-2-1 for different values of $I$, FedDA-2-1 outperforms Local-AMSGrad for all $I$ and with a greater margin for larger $I$. Note both Local-AMSGrad and FedDA-2-1 use Adam-style adaptive gradients (equation 6 and equation 7) and have same communication cost per epoch. In Figure 10, we compare FedAdam and Local-Adapt with FedDA-2-1. All methods use Adam-style adaptive gradients. FedAdam only performs adaptive gradients over the server, Local-Adapt performs both local and global adaptive gradients, but the state of the local adaptive gradient is refreshed per epoch. We have two observations: First, the Local-Adapt method has very marginal improvement over FedAdam, which shows the restarted strategy used by Local-Adapt is less effective than our method; Second, both FedAdam and Local-Adapt benefit little from increasing the $I$ value (compared to our FedDA-2-1). For FedAdam, this shows the limitation of only applying adaptive gradients at the server level. Finally, in Figure 11, we change $I$ for all four variants of our FedDA. As shown by the figure, our framework can benefit from more local steps.

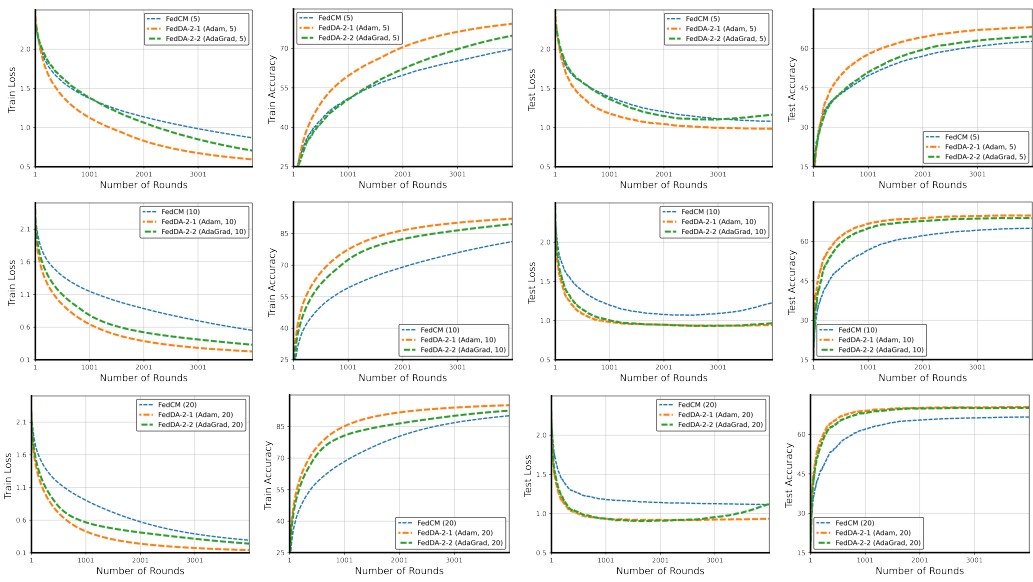

Figure 8: Comparison between FedCM vs FedDA-2-1 and FedDA-2-2. From top to bottom, we show $I = 5, 10, 20$ respectively. The number inside the parentheses is the value of $I$.

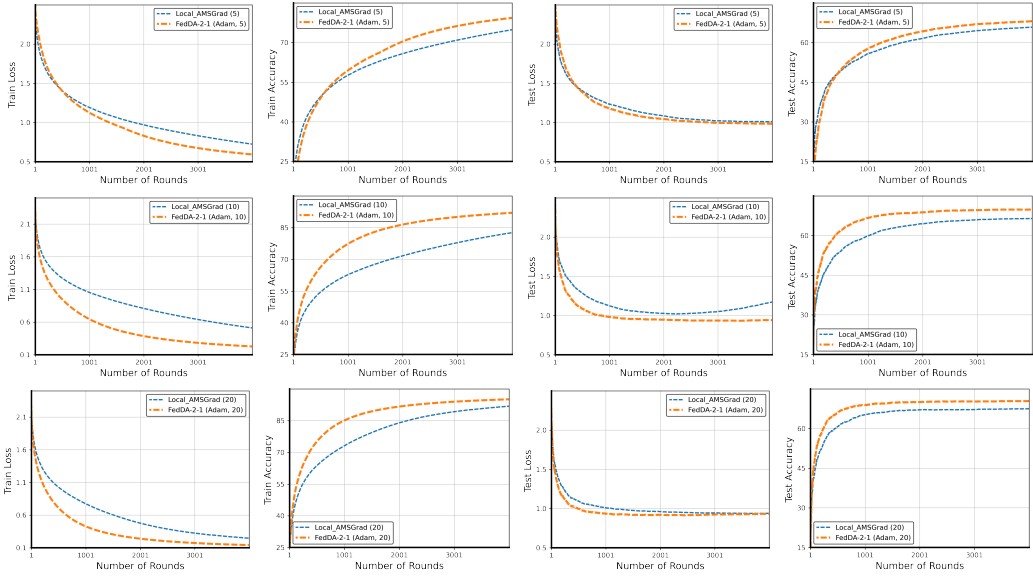

Figure 9: Comparison between Local-AMSGrad vs FedDA-2-1. From top to bottom, we show $I = 5, 10, 20$ respectively. The number inside the parentheses is the value of $I$.

### A.3   IMAGE CLASSIFICATION TASK WITH HETEROGENEOUS CIFAR10

For the heterogeneous case, we create heterogeneity in the training set as follows: Suppose we have 10 clients, for $i_{th}$ client, we distribute $\rho$-percent samples of $i_{th}$ class, and $(1 - \rho)/9$-percent samples of other classes, where $0 < \rho \leq 1$. Note for $\rho$ close to 1, the $i_{th}$ client will be dominated by images of $i_{th}$ class, thus the data distribution among clients will be very different. In our experiments, we choose $\rho = 0.8$. This means the $i_{th}$ client has 4000 images of $i_{th}$ class and 111 images of other classes. This creates a high level of heterogeneity.

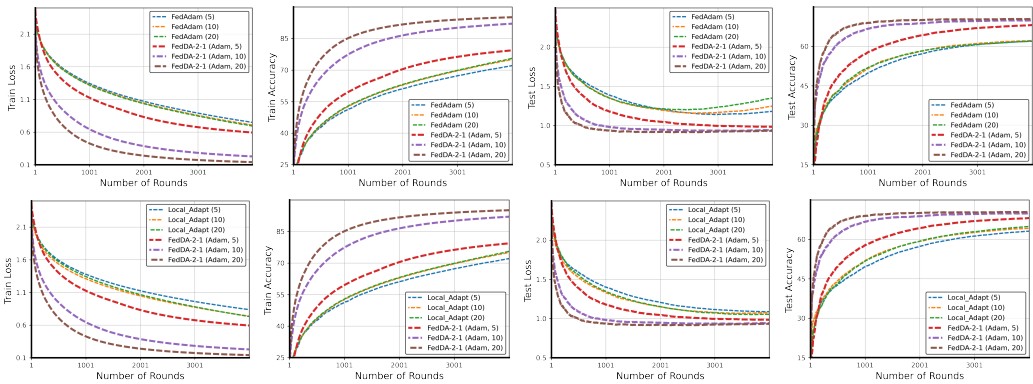

Figure 10: Comparison between FedAdam and Local-Adapt vs FedDA-2-1. The number inside the parentheses is the value of $I$.

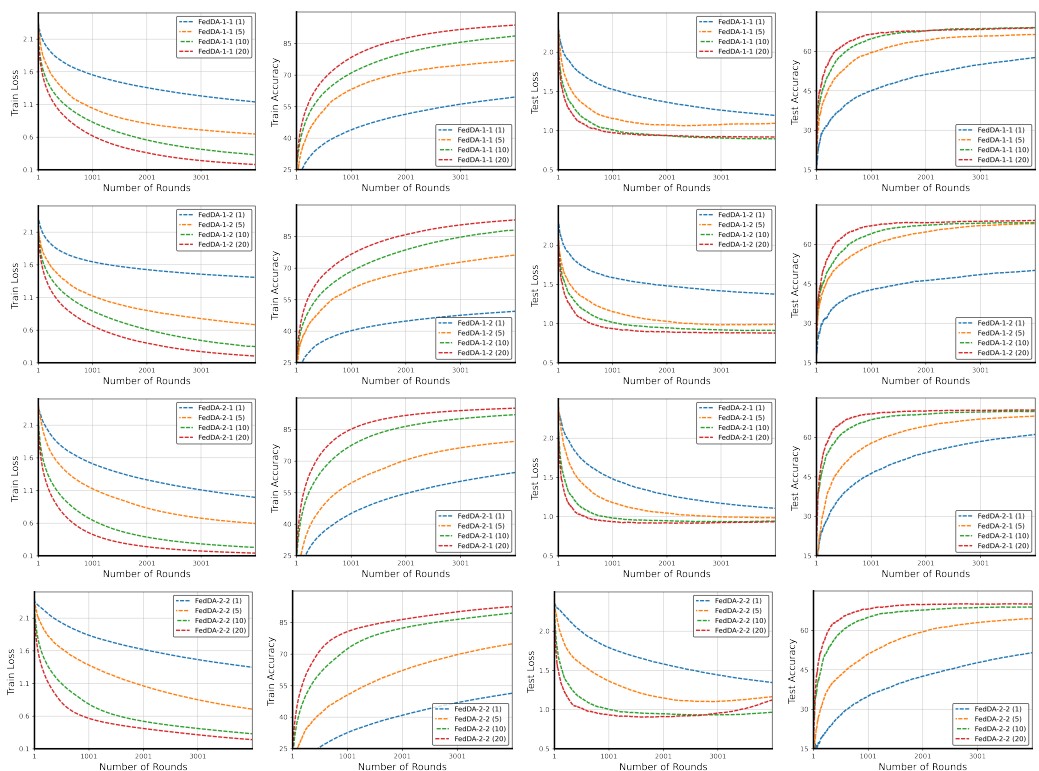

Figure 11: Ablation study of local steps $I$. From top row to the bottom row, we show results for FedDA-1-1, FedDA-1-2, FedDA-2-1 and FedDA-2-2. The number inside the parentheses is the value of $I$.

For hyper-parameters, we perform grid search and choose the best setting for each method. For the SGD method, we use learning rate 0.01; for the FedCM algorithm, we use learning rate 0.01, momentum coefficient $\alpha$ as 0.9; for the FedAdam algorithm, we use local learning rate 0.001, global learning rate 0.002, momentum coefficient 0.9, coefficient for adaptive matrix $\beta$ as 0.999; for the Local-Adapt algorithm, we use local learning rate 0.001, global learning rate 0.002, momentum coefficient 0.9, coefficient for adaptive matrix $\beta$ as 0.999; for the Local-AMSGrad algorithm, we use learning rate 0.001, momentum coefficient 0.9, adaptive matrix coefficient 0.999; for the MIME-MVR algorithm, we use learning rate 0.1, $w$ 100, $c$ as 2000; for the STEM algorithm, we use learning rate

Figure 12: Results for heterogeneous CIFAR10 dataset. From left to right, we show Train Loss, Train Accuracy, Test Loss, Test Accuracy *w.r.t* the number of rounds (E in Algorithm 1), respectively. $I$ is chosen as 5.

0.1, $w$ 100 and $c$ 2000; for our FedDA-MVR/FedDA-1-1, we use learning rate 0.02, $w$ as 10000, $c$ as 1000000, $\beta$ as 0.999 and $\tau$ as 0.01. For other variants of FedDA: for FedDA-2-1, we use learning rate 0.001, $\alpha$ as 0.9, $\beta$ as 0.999, $\tau$ as 0.01; for FedDA-1-2, we use learning rate 1, $w$ as 5000, $c$ as 100, $\beta$ as 0.999, $\tau$ as 0.01; for FedDA-2-2, we use learning rate 0.01, $\alpha$ 0.9, $\beta$ as 0.999, $\tau$ as 0.01.

## B  PROOF OF THEOREMS

In this section, we provide the convergence analysis of our algorithm.

### B.1  PRELIMINARY PROPOSITIONS

**Proposition B.1.** *Let $\{\theta_k\}, k \in K$ be $K$ vectors. Then the following are true: $||\theta_i + \theta_j||^2 \leq (1 + \lambda)||\theta_i||^2 + (1 + \frac{1}{\lambda})||\theta_j||^2$ for any $a > 0$ and $||\sum_{k=1}^{K} \theta_k||^2 \leq K \sum_{k=1}^{K} ||\theta_k||^2$*

**Proposition B.2.** *For a finite sequence $z^{(k)} \in \mathbb{R}^d$ for $k \in [K]$ define $\bar{z} := \frac{1}{K} \sum_{k=1}^{K} z^{(k)}$, we then have $\sum_{k=1}^{K} ||z^{(k)} - \bar{z}||^2 \leq \sum_{k=1}^{K} ||z^{(k)}||^2$.*

**Proposition B.3.** *Let $z_0 > 0$ and $z_1, z_2, \ldots, z_T \geq 0$. We have $\sum_{t=1}^{T} \frac{z_t}{z_0 + \sum_{i=t}^{t} z_i} \leq \log(1 + \frac{\sum_{i=1}^{t} z_i}{z_0})$.*

These propositions are standard results. For proofs, the reader can refer to Lemma 3 of Karimireddy et al. (2019a) for Proposition 1 and Lemma C.1 and Lemma C.2 in Khanduri et al. (2021a) for Propositions 2 and 3.

### B.2  PRELIMINARY LEMMAS IN LOCAL UPDATES

We first introduce some notation. For $0 \leq i \leq I$, we denote:

$$\psi_{\tau,i}^{(k)}(x) = -\langle x, z_{\tau,i}^{(k)} \rangle + \frac{1}{2}(x - x_{\tau,0}^{(k)})^T H_{\tau-1}(x - x_{\tau,0}^{(k)}), \tag{10}$$

then, by definition (Line 4 of Algorithm 2), we have:

$$x_{\tau,i}^{(k)} = \arg\min_{x \in \mathcal{X}} \psi_{\tau,i}^{(k)}(x), \tag{11}$$

we also define

$$\tilde{\psi}_{\tau,i}(x) = -\langle x, \bar{z}_{\tau,i} \rangle + \frac{1}{2}(x - x_{\tau,0})^T H_{\tau-1}(x - x_{\tau,0}), \tag{12}$$

where $\bar{z}_{\tau,i} = \frac{1}{K} \sum_{k=1}^{K} z_{\tau,i}^{(k)}$ is the virtual average of $z_{\tau,i}^{(k)}$ and $x_{\tau,0} = x_\tau$. Then we define

$$\tilde{x}_{\tau,i} = \arg\min_{x \in \mathcal{X}} \tilde{\psi}_{\tau,i}(x), \tag{13}$$

*Remark* B.4. Note that the global primal state $\tilde{x}_i$ is not the arithmetic mean of the local states $x_i^{(k)}$ in general.

Finally, we also define

$$\tilde{d}_{\tau,i} = \frac{1}{\eta_{\tau,i}}(\tilde{x}_{\tau,i} - \tilde{x}_{\tau,i+1}), \ d_{\tau,i}^{(k)} = \frac{1}{\eta_{\tau,i}}(x_{\tau,i}^{(k)} - x_{\tau,i+1}^{(k)}), k \in [K], i \in [I], \tag{14}$$

Furthermore, recall that by the procedure of Algorithm 2 (line 6), we have

$$\bar{\nu}_{\tau,i} = \frac{1}{\eta_{\tau,i}}(\bar{z}_{\tau,i} - \bar{z}_{\tau,i+1}), \ \nu_{\tau,i}^{(k)} = \frac{1}{\eta_{\tau,i}}(z_{\tau,i}^{(k)} - z_{\tau,i+1}^{(k)}), \ k \in [K], \ i \in [I], \tag{15}$$

*Remark* B.5. When it is clear from the context, we omit the global epoch $\tau$ in the subscript of the definitions, *i.e.* we use $\psi_i^{(k)}(x), \tilde{\psi}_i(x), x_i^{(k)}, \tilde{x}_i, \tilde{d}_i, d_i^{(k)}, \bar{\nu}_i, \nu_i^{(k)}$ and $H$.

Next, we introduce the following lemma related to local updates. We omit the global epoch number $\tau$ in the subscript.

**Lemma B.6.** *For any $i \in [I]$ and $k \in [K]$, we have the following inequalities be satisfied:*

1. *$\langle \nu_i^{(k)}, d_i^{(k)} \rangle \geq \rho||d_i^{(k)}||^2, ||\nu_i^{(k)}|| \geq \rho||d_i^{(k)}||$*

2. *$\langle \bar{\nu}_i, \tilde{d}_i \rangle \geq \rho||\tilde{d}_i||^2, ||\bar{\nu}_i|| \geq \rho||\tilde{d}_i||$;*

3. $||z_i^{(k)} - \bar{z}_i|| \geq \rho ||x_i^{(k)} - \tilde{x}_i||$;

*Proof.* The first and second claims follow similar derivations, and we provide only the derivations for the first claim. First, if $i = 1$, we have

$$x_1^{(k)} = \arg\min_{x \in \mathcal{X}} -\langle x, z_1^{(k)} \rangle + \frac{1}{2}(x - x_0^{(k)})^T H (x - x_0^{(k)}),$$

by the first-order optimality condition, we have:

$$\langle -z_1^{(k)} + H(x_1^{(k)} - x_0^{(k)}), u - x_1^{(k)} \rangle \geq 0, \ \forall \, u \in \mathcal{X},$$

choose $u = x_0^{(k)}$ and use the fact that $z_1^{(k)} = -\eta_0 \nu_0$, we have:

$$\eta_0 ||\nu_0^{(k)}|| \times ||x_0^{(k)} - x_1^{(k)}|| \geq \eta_0 \langle \nu_0^{(k)}, x_0^{(k)} - x_1^{(k)} \rangle \geq (x_1^{(k)} - x_0^{(k)})^T H (x_1^{(k)} - x_0^{(k)}) \geq \rho ||x_0^{(k)} - x_1^{(k)}||^2$$

we use the Cauchy-Schwartz inequality in the leftmost inequality and use the strong convexity assumption of the adaptive matrix in the rightmost inequality, we get the result in the lemma.

Next if $i > 0$, by the definition of $\psi_i^{(k)}(x)$, we have:

$$\psi_i^{(k)}(x_{i+1}^{(k)}) - \psi_i^{(k)}(x_i^{(k)}) = -\langle z_i^{(k)}, x_{i+1}^{(k)} - x_i^{(k)} \rangle + \frac{1}{2}(x_{i+1}^{(k)} - x_i^{(k)})^T H (x_{i+1}^{(k)} + x_i^{(k)} - 2x_0^{(k)}) \tag{16}$$

Then by the definition of $x_i^{(k)}$, and the first order optimality condition, we have

$$\langle -z_i^{(k)} + H(x_i^{(k)} - x_0^{(k)}), u - x_i^{(k)} \rangle \geq 0, \ \forall \, u \in \mathcal{X},$$

if we pick $u = x_{i+1}^{(k)}$, we have $-\langle z_i^{(k)}, x_{i+1}^{(k)} - x_i^{(k)} \rangle \geq -(x_{i+1}^{(k)} - x_i^{(k)})^T H (x_i^{(k)} - x_0^{(k)})$, plug this inequality to equation 16, we have:

$$\psi_i^{(k)}(x_{i+1}^{(k)}) - \psi_i^{(k)}(x_i^{(k)})$$
$$\geq -(x_{i+1}^{(k)} - x_i^{(k)})^T H (x_i^{(k)} - x_0^{(k)}) + \frac{1}{2}(x_{i+1}^{(k)} - x_i^{(k)})^T H (x_{i+1}^{(k)} + x_i^{(k)} - 2x_0^{(k)})$$
$$\geq \frac{1}{2}(x_{i+1}^{(k)} - x_i^{(k)})^T H (x_{i+1}^{(k)} - x_i^{(k)})$$

Similarly for $\psi_{i+1}^{(k)}$, we have:

$$\psi_{i+1}^{(k)}(x_{i+1}^{(k)}) - \psi_{i+1}^{(k)}(x_i^{(k)}) = -\langle z_{i+1}^{(k)}, x_{i+1}^{(k)} - x_i^{(k)} \rangle + \frac{1}{2}(x_{i+1}^{(k)} - x_i^{(k)})^T H (x_{i+1}^{(k)} + x_i^{(k)} - 2x_0^{(k)})$$

and by the definition of $x_{i+1}^{(k)}$ and the first order optimality condition, we can get

$$\langle -z_{i+1}^{(k)} + H(x_{i+1}^{(k)} - x_0^{(k)}), u - x_{i+1}^{(k)} \rangle \geq 0, \ \forall \, u \in \mathcal{X},$$

pick $u = x_i^{(k)}$, we have $-\langle z_{i+1}^{(k)}, x_{i+1}^{(k)} - x_i^{(k)} \rangle \leq -(x_{i+1}^{(k)} - x_i^{(k)})^T H (x_{i+1}^{(k)} - x_0^{(k)})$, plug this inequality to the above equality, we have:

$$\psi_{i+1}^{(k)}(x_{i+1}^{(k)}) - \psi_{i+1}^{(k)}(x_i^{(k)})$$
$$\leq -(x_{i+1}^{(k)} - x_i^{(k)})^T H (x_{i+1}^{(k)} - x_0^{(k)}) + \frac{1}{2}(x_{i+1}^{(k)} - x_i^{(k)})^T H (x_{i+1}^{(k)} + x_i^{(k)} - 2x_0^{(k)})$$
$$\leq -\frac{1}{2}(x_{i+1}^{(k)} - x_i^{(k)})^T H (x_{i+1}^{(k)} - x_i^{(k)})$$

Next, by definition of $\psi_i^{(k)}(x)$ and $\psi_{i+1}^{(k)}(x)$, we have:

$$\psi_{i+1}^{(k)}(x_{i+1}^{(k)}) - \psi_{i+1}^{(k)}(x_i^{(k)}) = \psi_i^{(k)}(x_{i+1}^{(k)}) - \psi_i^{(k)}(x_i^{(k)}) + \eta_i \langle \nu_i^{(k)}, x_{i+1}^{(k)} - x_i^{(k)} \rangle$$

Finally, we combine the above relations and have:

$$\eta_i ||\nu_i^{(k)}|| \times ||x_i^{(k)} - x_{i+1}^{(k)}|| \geq \eta_i \langle \nu_i^{(k)}, x_i^{(k)} - x_{i+1}^{(k)} \rangle \geq (x_{i+1}^{(k)} - x_i^{(k)})^T H (x_{i+1}^{(k)} - x_i^{(k)}) \geq \rho ||x_i^{(k)} - x_{i+1}^{(k)}||^2$$

we use the Cauchy-Schwartz inequality in the leftmost inequality and use the strong convexity assumption of the adaptive matrix in the rightmost inequality, we get the result in the claim of the lemma.

Next, we prove the third claim, by the definition of $\psi_{i+1}^{(k)}$, we have:

$$\psi_{i+1}^{(k)}(x_{i+1}^{(k)}) - \psi_{i+1}^{(k)}(\tilde{x}_{i+1}) = -\langle z_{i+1}^{(k)}, x_{i+1}^{(k)} - \tilde{x}_{i+1} \rangle + \frac{1}{2}(x_{i+1}^{(k)} - \tilde{x}_{i+1})^T H(x_{i+1}^{(k)} + \tilde{x}_{i+1} - 2x_0^{(k)})$$

By the definition of $x_{i+1}^{(k)}$ and first order optimality condition, we have

$$\langle -z_{i+1}^{(k)} + H(x_{i+1}^{(k)} - x_0^{(k)}), u - x_{i+1}^{(k)} \rangle \geq 0, \ \forall \ u \in \mathcal{X},$$

pick $u = \tilde{x}_{i+1}$, we have $-\langle z_{i+1}^{(k)}, x_{i+1}^{(k)} - \tilde{x}_{i+1} \rangle \leq -(x_{i+1}^{(k)} - \tilde{x}_{i+1})^T H(x_{i+1}^{(k)} - x_0^{(k)})$. Plug this inequality back to the above inequality, we have:

$$\psi_{i+1}^{(k)}(x_{i+1}^{(k)}) - \psi_{i+1}^{(k)}(\tilde{x}_{i+1})$$
$$\leq -(x_{i+1}^{(k)} - \tilde{x}_{i+1})^T H(x_{i+1}^{(k)} - x_0^{(k)}) + \frac{1}{2}(x_{i+1}^{(k)} - \tilde{x}_{i+1})^T H(x_{i+1}^{(k)} + \tilde{x}_{i+1} - 2x_0^{(k)})$$
$$\leq -\frac{1}{2}(x_{i+1}^{(k)} - \tilde{x}_{i+1})^T H(x_{i+1}^{(k)} - \tilde{x}_{i+1})$$

Then for $\tilde{\psi}_{i+1}(x)$, we have:

$$\tilde{\psi}_{i+1}^{(k)}(x_{i+1}^{(k)}) - \tilde{\psi}_{i+1}(\tilde{x}_{i+1}) = -\langle \bar{z}_{i+1}, x_{i+1}^{(k)} - \tilde{x}_{i+1} \rangle + \frac{1}{2}(x_{i+1}^{(k)} - \tilde{x}_{i+1})^T H(x_{i+1}^{(k)} + \tilde{x}_{i+1} - 2\tilde{x}_0)$$

By the definition of $\tilde{x}_{i+1}$ and first order optimality condition, we have:

$$\langle -\bar{z}_{i+1} + H(\tilde{x}_{i+1} - \tilde{x}_0), u - \tilde{x}_{i+1} \rangle \geq 0, \ \forall \ u \in \mathcal{X},$$

pick $u = x_{i+1}^{(k)}$, we have $-\langle \bar{z}_{i+1}, x_{i+1}^{(k)} - \tilde{x}_{i+1} \rangle \geq -(x_{i+1}^{(k)} - \tilde{x}_{i+1})^T H(\tilde{x}_{i+1} - \tilde{x}_0)$. Plug this inequality back to the above inequality, we have:

$$\tilde{\psi}_{i+1}(x_{i+1}^{(k)}) - \tilde{\psi}_{i+1}(\tilde{x}_{i+1})$$
$$\geq -(x_{i+1}^{(k)} - \tilde{x}_{i+1})^T H(\tilde{x}_{i+1} - \tilde{x}_0) + \frac{1}{2}(x_{i+1}^{(k)} - \tilde{x}_{i+1})^T H(x_{i+1}^{(k)} + \tilde{x}_{i+1} - 2\tilde{x}_0)$$
$$\geq \frac{1}{2}(x_{i+1}^{(k)} - \tilde{x}_{i+1})^T H(x_{i+1}^{(k)} - \tilde{x}_{i+1})$$

Next, since we have $x_0^{(k)} = \tilde{x}_0$, then by the definition of $\psi_{i+1}^{(k)}(x)$ and $\tilde{\psi}_{i+1}(x)$ we have:

$$\psi_{i+1}^{(k)}(x_{i+1}^{(k)}) - \psi_{i+1}^{(k)}(\tilde{x}_{i+1}) = \tilde{\psi}_{i+1}(x_{i+1}^{(k)}) - \tilde{\psi}_{i+1}(\tilde{x}_{i+1}) - \langle z_{i+1}^{(k)} - \bar{z}_{i+1}, x_{i+1}^{(k)} - \tilde{x}_{i+1} \rangle$$

Next, we combine the above relations and have:

$$||z_{i+1}^{(k)} - \bar{z}_{i+1}|| \times ||x_{i+1}^{(k)} - \tilde{x}_{i+1}|| \geq \langle z_{i+!}^{(k)} - \bar{z}_{i+1}, x_{i+1}^{(k)} - \tilde{x}_{i+1} \rangle$$
$$\geq (x_{i+1}^{(k)} - \tilde{x}_{i+1})^T H(x_{i+1}^{(k)} - \tilde{x}_{i+1}) \geq \rho ||x_{i+1}^{(k)} - \tilde{x}_{i+1}||^2$$

where the first inequality is by the Cauchy-Schwartz inequality and the last inequality is by the positive definiteness of $H$. This concludes the proof of the first inequality in the lemma. $\qquad \square$

### B.3 STATE CONSENSUS ERROR

As each client performs local update, the states *i.e.* $z_{\tau,i}^{(k)}$ and $\nu_{\tau,i}^{(k)}$ drift away, the following lemmas bound this difference. We omit the global epoch number $\tau$ in the subscript.

**Lemma B.7.** *For each $0 \leq i \leq I$, and suppose iterates $z_i^{(k)}$, $k \in [K]$ are generated from Algorithm 2, we have:*

$$\sum_{k=1}^{K} \mathbb{E}\|z_i^{(k)} - \bar{z}_i\|^2 \leq (I-1) \sum_{\ell=1}^{i-1} \eta_\ell^2 \sum_{k=1}^{K} \mathbb{E}\|\nu_\ell^{(k)} - \bar{\nu}_\ell\|^2,$$

*where the expectation is w.r.t the stochasticity of the algorithm.*

*Proof.* Based on Algorithm 2, we have $z_0^{(k)} = \bar{z}_0 = 0$, the inequality in the lemma holds trivially. Otherwise, we have

$$z_i^{(k)} = -\sum_{\ell=0}^{i-1} \eta_\ell \nu_\ell^{(k)} \quad \text{and} \quad \bar{z}_i = -\sum_{\ell=0}^{i-1} \eta_\ell \bar{\nu}_\ell.$$

So we have:

$$\sum_{k=1}^{K} \|z_i^{(k)} - \bar{z}_i\|^2 = \sum_{k=1}^{K} \Big\| \sum_{\ell=1}^{i-1} \big(\eta_\ell \nu_\ell^{(k)} - \eta_\ell \bar{\nu}_\ell\big) \Big\|^2 \leq (I-1) \sum_{\ell=1}^{i-1} \eta_\ell^2 \sum_{k=1}^{K} \|\nu_\ell^{(k)} - \bar{\nu}_\ell\|^2$$

where the equality uses the fact $\nu_0^{(k)} = \nu_0$ for $k \in [K]$, the inequality uses the Proposition B.1 and the fact that we have $i \leq I$. We get the claim in the lemma by taking expectation on both sides of the above inequality. This completes the proof. □

**Lemma B.8.** *For $i \in [I]$, we have:*

$$\sum_{k=1}^{K} \|d_i^{(k)} - \tilde{d}_i\|^2 \leq \frac{4^2(I-1)}{\rho^2 \eta_i^2} \sum_{\ell=1}^{i} \eta_\ell^2 \sum_{k=1}^{K} \mathbb{E}\|\nu_\ell^{(k)} - \bar{\nu}_\ell\|^2$$

*where the expectation is w.r.t the stochasticity of the algorithm.*

*Proof.* Firstly, when $i = 0$, $x_0^{(k)} = \tilde{x}_0$, $z_1^{(k)} = \bar{z}_1$, so we have $x_1^{(k)} = \tilde{x}_1$ by Line 5 of Algorithm 2, and then we have $\eta_0 d_0^{(k)} = x_0^{(k)} - x_1^{(k)} = \tilde{x}_0 - \tilde{x}_1 = \eta_t \tilde{d}_0$, the inequality in the lemma holds trivially.

Next when $i > 0$, we have:

$$\eta_i^2 \|d_i^{(k)} - \tilde{d}_i\|^2 = \|x_i^{(k)} - x_{i+1}^{(k)} - (\tilde{x}_i - \tilde{x}_{i+1})\|^2 \leq 2\|x_i^{(k)} - \tilde{x}_i\|^2 + 2\|x_{i+1}^{(k)} - \tilde{x}_{i+1}\|^2$$

$$\leq \frac{2^2}{\rho^2} \big( \|z_i^{(k)} - \bar{z}_i\|^2 + \|z_{i+1}^{(k)} - \bar{z}_{i+1}\|^2 \big)$$

The last inequality uses claim 3 of Lemma B.6. Sum over $k \in [K]$ and use Lemma B.7, we have:

$$\rho^2 \eta_i^2 \sum_{k=1}^{K} \|d_i^{(k)} - \tilde{d}_i\|^2 \leq 2^2(I-1) \sum_{\ell=1}^{i-1} \eta_\ell^2 \sum_{k=1}^{K} \mathbb{E}\|\nu_\ell^{(k)} - \bar{\nu}_\ell\|^2 + 2^2(I-1) \sum_{\ell=1}^{i} \eta_\ell^2 \sum_{k=1}^{K} \mathbb{E}\|\nu_\ell^{(k)} - \bar{\nu}_\ell\|^2$$

$$\leq 4^2(I-1) \sum_{\ell=1}^{i} \eta_\ell^2 \sum_{k=1}^{K} \mathbb{E}\|\nu_\ell^{(k)} - \bar{\nu}_\ell\|^2$$

This completes the proof. □

## B.4 DESCENT LEMMA

In this subsection, we bound the descent of function value $f(\tilde{x}_{\tau,i})$ over the virtual sequence $\tilde{x}_{\tau,i}$.

**Lemma B.9.** *Suppose that the sequence $\{x_{\tau,i}^{(k)}\}_{i=0}^{I-1}$ be generated from Algorithm 2, then we have*

$$f(\tilde{x}_{\tau+1}) \leq f(\tilde{x}_{\tau}) - \sum_{i=0}^{I-1} \left( \frac{3\rho\eta_{\tau+1,i}}{4} - \frac{\eta_{\tau+1,i}^2 L}{2} \right) \|\tilde{d}_{\tau+1,i}\|^2 + \sum_{i=0}^{I-1} \frac{\eta_{\tau+1,i}}{\rho} \|\bar{e}_{\tau+1,i}\|^2$$

*where $\bar{e}_{\tau,i} = \bar{\nu}_{\tau,i} - \frac{1}{K}\sum_{k=1}^{K} \nabla f^{(k)}(x_{\tau,i}^{(k)})$.*

*Proof.* Since the function $f(x)$ is $L$-smooth, we have (we omit the global epoch number $\tau$ for ease of notation):

$$f(\tilde{x}_{i+1}) \leq f(\tilde{x}_i) + \langle \nabla f(\tilde{x}_i), \tilde{x}_{i+1} - \tilde{x}_i \rangle + \frac{L}{2}\|\tilde{x}_{i+1} - \tilde{x}_i\|^2 = f(\tilde{x}_i) - \eta_i \langle \nabla f(\tilde{x}_i), \tilde{d}_i \rangle + \frac{L\eta_i^2}{2}\|\tilde{d}_i\|^2$$

$$= f(\tilde{x}_i) - \eta_i \langle \bar{\nu}_i, \tilde{d}_i \rangle - \eta_i \langle \nabla f(\tilde{x}_i) - \bar{\nu}_i, \tilde{d}_i \rangle + \frac{L\eta_i^2}{2}\|\tilde{d}_i\|^2$$

$$\overset{(a)}{\leq} f(\tilde{x}_i) - (\rho\eta_i - \frac{L\eta_i^2}{2})\|\tilde{d}_i\|^2 - \eta_i \langle \nabla f(\tilde{x}_i) - \bar{\nu}_i, \tilde{d}_i \rangle$$

$$\overset{(b)}{\leq} f(\tilde{x}_i) - \left( \rho\eta_i - \frac{\eta_i^2 L}{2} \right) \|\tilde{d}_i\|^2 + \frac{\rho\eta_i}{4}\|\tilde{d}_i\|^2 + \frac{\eta_i}{\rho}\|\bar{\nu}_i - \nabla f(\tilde{x}_i)\|^2$$

$$\overset{(c)}{\leq} f(\tilde{x}_i) - \left( \frac{3\rho\eta_i}{4} - \frac{\eta_i^2 L}{2} \right) \|\tilde{d}_i\|^2 + \frac{\eta_i}{\rho}\|\bar{e}_i\|^2$$

In inequality (a), we use claim 1 of Lemma B.6; inequality (b) uses Young's inequality; inequality (c) denotes $\bar{e}_i = \bar{\nu}_i - \nabla f(\tilde{x}_i)$. For $\tilde{e}_i$, we have: For the $\tau$ global epoch, we sum over $i = 0$ to $I-1$, we have:

$$f(\tilde{x}_{\tau+1,I}) \leq f(\tilde{x}_{\tau+1,0}) - \sum_{i=0}^{I-1} \left( \frac{3\rho\eta_{\tau+1,i}}{4} - \frac{\eta_{\tau+1,i}^2 L}{2} \right) \|\tilde{d}_{\tau+1,i}\|^2 + \sum_{i=0}^{I-1} \frac{2\eta_{\tau+1,i}}{\rho} \|\bar{e}_{\tau+1,i}\|^2$$

Follow the update rules in Algorithm 1 and Algorithm 2, we have $\tilde{x}_{\tau+1,0} = x_{\tau}$ and $\tilde{x}_{\tau+1,I} = x_{\tau+1}$. This completes the proof. □

## B.5 GRADIENT ERROR CONTRACTION

In this subsection, we bound the gradient estimation error $\bar{e}_{\tau,i}$, where we have $\bar{e}_{\tau,i} = \bar{\nu}_{\tau,i} - \nabla f(\tilde{x}_{\tau,i})$ as defined in Lemma B.9, additionally, we also define the global gradient estimation error $e_{\tau}$ as $\bar{e}_{\tau} = \nu_{\tau} - \nabla f(x_{\tau})$. Note we have $e_{\tau} = \bar{e}_{\tau,I} = \bar{e}_{\tau+1,0}$. We first show a fact about $\bar{e}_0$, the initial gradient estimation error.

**Lemma B.10.** *For $e_0 := \nu_0 - \nabla f(x_0)$, suppose we choose mini-batch size of $|\mathcal{B}_0^{(k)}| = b_1, k \in [K]]$, we have: $\mathbb{E}\|e_0\|^2 \leq \frac{\sigma^2}{b_1 K}$.*

*Proof.* By line 1 of Algorithm 1, we have:

$$\mathbb{E}\|e_0\|^2 = \mathbb{E}\left\| \nu_0 - \frac{1}{K}\sum_{k=1}^{K} \nabla f^{(k)}(x_0) \right\|^2$$

$$= \mathbb{E}\left\| \frac{1}{K}\sum_{k=1}^{K} \nabla f^{(k)}(x_0; \mathcal{B}_0^{(k)}) - \frac{1}{K}\sum_{k=1}^{K} \nabla f^{(k)}(x_0) \right\|^2$$

$$\overset{(a)}{\leq} \frac{1}{K^2}\sum_{k=1}^{K} \mathbb{E}\left\| \nabla f^{(k)}(x_0; \mathcal{B}_0^{(k)}) - \nabla f^{(k)}(x_0) \right\|^2 \overset{(b)}{\leq} \frac{\sigma^2}{b_1 K}.$$

where $(a)$ follows from the following: From the unbiased gradient assumption, we have: $\mathbb{E}\big[\nabla f^{(k)}(x_0^{(k)}; \mathcal{B}_0^{(k)})\big] = \nabla f^{(k)}(x_0^{(k)})$, for all $k \in [K]$. Moreover, the samples $\mathcal{B}_0^{(k)}$ and $\mathcal{B}_0^{(\ell)}$ at

the $k^{\text{th}}$ and the $\ell^{\text{th}}$ clients are chosen uniformly randomly, and independent of each other for all $k, \ell \in [K]]$ and $k \neq \ell$. Inequality $(b)$ results from the bounded variance assumption. This completes the proof. $\qquad\square$

**Lemma B.11.** *Define* $\bar{e}_{\tau,i} := \bar{\nu}_{\tau,i} - \frac{1}{K} \sum_{k=1}^{K} \nabla f^{(k)}(\tilde{x}_{\tau,i})$, *then for every* $\tau \geq 1$ *and* $i \geq 1$, *suppose* $\alpha_i < 1$ *and clients use batchsize* $b$ *in the training, then we have:*

$$
\mathbb{E}\|\bar{e}_{\tau,i}\|^2 \leq (1 - \alpha_{\tau,i})^2 \mathbb{E}\|\bar{e}_{\tau,i-1}\|^2 + \frac{256(I-1)L^2}{\rho^2 K} \sum_{\ell=1}^{i-1} \eta_{\tau,\ell}^2 \sum_{k=1}^{K} \mathbb{E}\|\nu_{\tau,\ell}^{(k)} - \bar{\nu}_{\tau,\ell}\|^2
$$
$$
+ \frac{8\eta_{\tau,i-1}^2 L^2}{K} \mathbb{E}\|\tilde{d}_{\tau,i-1}\|^2 + \frac{4\alpha_{\tau,i}^2 \sigma^2}{bK}
$$

*where the expectation is w.r.t the stochasticity of the algorithm.*

*Proof.* Consider the error term $\|\bar{e}_i\|^2$, $i \geq 1$ (we omit the global epoch number $\tau$ for ease of notation), we have:

$$
\mathbb{E}\|\bar{e}_i\|^2 = \mathbb{E}\left\| \bar{\nu}_i - \frac{1}{K} \sum_{k=1}^{K} \nabla f^{(k)}(\tilde{x}_i) \right\|^2
$$

$$
= \mathbb{E}\left\| \frac{1}{K} \sum_{k=1}^{K} \nabla f^{(k)}(x_i^{(k)}; \mathcal{B}_i^{(k)}) + (1-\alpha_i)\left( \bar{\nu}_{i-1} - \frac{1}{K} \sum_{k=1}^{K} \nabla f^{(k)}(x_{i-1}^{(k)}; \mathcal{B}_i^{(k)}) \right) - \frac{1}{K} \sum_{k=1}^{K} \nabla f^{(k)}(\tilde{x}_i) \right\|^2
$$

$$
= \mathbb{E}\left\| \frac{1}{K} \sum_{k=1}^{K} \left( \left( \nabla f^{(k)}(x_i^{(k)}; \mathcal{B}_i^{(k)}) - \nabla f^{(k)}(\tilde{x}_i) \right) \right.\right.
$$
$$
\left.\left. - (1-\alpha_i)\left( \nabla f^{(k)}(x_{i-1}^{(k)}; \mathcal{B}_i^{(k)}) - \nabla f^{(k)}(\tilde{x}_{i-1}) \right) \right) + (1-\alpha_i)\bar{e}_{i-1} \right\|^2
$$

$$
= (1-\alpha_i)^2 \mathbb{E}\|\bar{e}_{i-1}\|^2 + \frac{1}{K^2} \mathbb{E}\left\| \sum_{k=1}^{K} \left[ \left( \nabla f^{(k)}(x_i^{(k)}; \mathcal{B}_i^{(k)}) - \nabla f^{(k)}(\tilde{x}_i) \right) \right.\right.
$$
$$
\left.\left. - (1-\alpha_i)\left( \nabla f^{(k)}(x_{i-1}^{(k)}; \mathcal{B}_i^{(k)}) - \nabla f^{(k)}(\tilde{x}_{i-1}) \right) \right] \right\|^2
$$

$$
\leq (1-\alpha_i)^2 \mathbb{E}\|\bar{e}_{i-1}\|^2 + \frac{2}{K^2} \mathbb{E}\left\| \sum_{k=1}^{K} \left[ \left( \nabla f^{(k)}(x_i^{(k)}; \mathcal{B}_i^{(k)}) - \nabla f^{(k)}(x_i^{(k)}) \right) \right.\right.
$$
$$
\left.\left. - (1-\alpha_i)\left( \nabla f^{(k)}(x_{i-1}^{(k)}; \mathcal{B}_i^{(k)}) - \nabla f^{(k)}(x_{i-1}^{(k-1)}) \right) \right] \right\|^2
$$

$$
+ \frac{2}{K^2} \mathbb{E}\left\| \sum_{k=1}^{K} \left[ \left( \nabla f^{(k)}(x_i^{(k)}) - \nabla f^{(k)}(\tilde{x}_i) \right) - (1-\alpha_i)\left( \nabla f^{(k)}(x_{i-1}^{(k)}) - \nabla f^{(k)}(\tilde{x}_{i-1}) \right) \right] \right\|^2
$$

$$
\leq (1-\alpha_i)^2 \mathbb{E}\|\bar{e}_{i-1}\|^2 + \frac{2}{K^2} \sum_{k=1}^{K} \mathbb{E}\left\| \left( \nabla f^{(k)}(x_i^{(k)}; \mathcal{B}_i^{(k)}) - \nabla f^{(k)}(x_i^{(k)}) \right) \right.
$$
$$
\left. - (1-\alpha_i)\left( \nabla f^{(k)}(x_{i-1}^{(k)}; \mathcal{B}_i^{(k)}) - \nabla f^{(k)}(x_{i-1}^{(k)}) \right) \right\|^2
$$

$$
+ \frac{2}{K} \sum_{k=1}^{K} \mathbb{E}\left\| \left( \nabla f^{(k)}(x_i^{(k)}) - \nabla f^{(k)}(\tilde{x}_i) \right) - (1-\alpha_i)\left( \nabla f^{(k)}(x_{i-1}^{(k)}) - \nabla f^{(k)}(\tilde{x}_{i-1}) \right) \right\|^2
$$

where the first equality uses the definition of $\bar{\nu}_i$; last equality follows from expanding the norm using the inner products across $k \in [K]$ and noting that the cross terms of the second term is zero in expectation because of the samples are sampled independently at different workers.

We first consider the second term above:

$$\mathbb{E}\big\|\big(\nabla f^{(k)}(x_i^{(k)};\mathcal{B}_i^{(k)}) - \nabla f^{(k)}(x_i^{(k)})\big) - (1-\alpha_i)\big(\nabla f^{(k)}(x_{i-1}^{(k)};\mathcal{B}_i^{(k)}) - \nabla f^{(k)}(x_{i-1}^{(k)})\big)\big\|^2$$

$$= \mathbb{E}\big\|(1-a_i)\big[\big(\nabla f^{(k)}(x_i^{(k)};\mathcal{B}_i^{(k)}) - \nabla f^{(k)}(x_i^{(k)})\big) - \big(\nabla f^{(k)}(x_{i-1}^{(k)};\mathcal{B}_i^{(k)}) - \nabla f^{(k)}(x_{i-1}^{(k)})\big)\big]$$
$$+ \alpha_i\big(\nabla f^{(k)}(x_i^{(k)};\mathcal{B}_i^{(k)}) - \nabla f^{(k)}(x_i^{(k)})\big)\big\|^2$$

$$\le 2(1-\alpha_i)^2\mathbb{E}\big\|\big(\nabla f^{(k)}(x_i^{(k)};\mathcal{B}_i^{(k)}) - \nabla f^{(k)}(x_{i-1}^{(k)};\mathcal{B}_i^{(k)})\big) - \big(\nabla f^{(k)}(x_i^{(k)}) - \nabla f^{(k)}(x_{i-1}^{(k)})\big)\big\|^2$$
$$+ 2\alpha_i^2\mathbb{E}\big\|\nabla f^{(k)}(x_i^{(k)};\mathcal{B}_i^{(k)}) - \nabla f^{(k)}(x_i^{(k)})\big\|^2$$

$$\le 2(1-\alpha_i)^2\mathbb{E}\big\|\nabla f^{(k)}(x_i^{(k)};\mathcal{B}_i^{(k)}) - \nabla f^{(k)}(x_{i-1}^{(k)};\mathcal{B}_i^{(k)})\big\|^2 + 2\alpha_i^2\sigma^2/b$$

$$\le 2(1-\alpha_i)^2 L^2\mathbb{E}\|x_i^{(k)} - x_{i-1}^{(k)}\|^2 + 2a_i^2\sigma^2/b \le 2(1-\alpha_i)^2 L^2\eta_{i-1}^2\mathbb{E}\|d_{i-1}^{(k)}\|^2 + 2\alpha_i^2\sigma^2/b$$

$$\le 4(1-\alpha_i)^2 L^2\eta_{i-1}^2\mathbb{E}\|d_{i-1}^{(k)} - \tilde{d}_{i-1}\|^2 + 4(1-\alpha_i)^2 L^2\eta_{i-1}^2\mathbb{E}\|\tilde{d}_{i-1}\|^2 + 2\alpha_i^2\sigma^2/b$$

where we use Proposition B.1 in the first inequality and the bounded variance assumption in the second inequality.

For the third term, we have:

$$\mathbb{E}\big\|\nabla f^{(k)}(x_i^{(k)}) - \nabla f^{(k)}(\tilde{x}_i) - (1-\alpha_i)\big(\nabla f^{(k)}(x_{i-1}^{(k)}) - \nabla f^{(k)}(\tilde{x}_{i-1})\big)\big\|^2$$

$$\overset{(a)}{\le} 2\mathbb{E}\big\|\nabla f^{(k)}(x_i^{(k)}) - \nabla f^{(k)}(\tilde{x}_i)\big\|^2 + 2\mathbb{E}\big\|(1-\alpha_i)\big(\nabla f^{(k)}(x_{i-1}^{(k)}) - \nabla f^{(k)}(\tilde{x}_{i-1})\big)\big\|^2$$

$$\le 2L^2\mathbb{E}\big\|x_i^{(k)} - \tilde{x}_i\big\|^2 + 2L^2(1-\alpha_i)^2\mathbb{E}\big\|x_{i-1}^{(k)} - \tilde{x}_{i-1}\big\|^2$$

$$\overset{(b)}{\le} \frac{2L^2}{\rho^2}\mathbb{E}\big\|z_i^{(k)} - \bar{z}_i\big\|^2 + \frac{2L^2(1-\alpha_i)^2}{\rho^2}\mathbb{E}\big\|z_{i-1}^{(k)} - \bar{z}_{i-1}\big\|^2$$

where (a) uses Proposition B.1; (b) uses claim 3 of Lemma B.6;

Finally, we combine the above inequalities together to get:

$$\mathbb{E}\|\bar{e}_i\|^2 \le (1-\alpha_i)^2\mathbb{E}\|\bar{e}_{i-1}\|^2 + \frac{4\alpha_i^2\sigma^2}{bK} + \frac{8\eta_{i-1}^2 L^2}{K^2}\sum_{k=1}^K \mathbb{E}\|d_{i-1}^{(k)} - \tilde{d}_{i-1}\|^2$$

$$+ \frac{8\eta_{i-1}^2 L^2}{K}\mathbb{E}\|\tilde{d}_{i-1}\|^2 + \frac{8L^2}{K\rho^2}\sum_{k=1}^K\left[\mathbb{E}\big\|z_i^{(k)} - \bar{z}_i\big\|^2 + \mathbb{E}\big\|z_{i-1}^{(k)} - \bar{z}_{i-1}\big\|^2\right]$$

$$\le (1-\alpha_i)^2\mathbb{E}\|\bar{e}_{i-1}\|^2 + \frac{4\alpha_i^2\sigma^2}{bK} + \frac{128(I-1)L^2}{K^2\rho^2}\sum_{\ell=1}^{i-1}\eta_\ell^2\sum_{k=1}^K\mathbb{E}\|\nu_\ell^{(k)} - \bar{\nu}_\ell\|^2$$

$$+ \frac{16\eta_{i-1}^2 L^2}{K}\mathbb{E}\|\tilde{d}_{i-1}\|^2 + \frac{16(I-1)L^2}{K\rho^2}\sum_{\ell=1}^{i-1}\eta_\ell^2\sum_{k=1}^K\mathbb{E}\|\nu_\ell^{(k)} - \bar{\nu}_\ell\|^2$$

$$\le (1-\alpha_i)^2\mathbb{E}\|\bar{e}_{i-1}\|^2 + \frac{4\alpha_i^2\sigma^2}{bK}$$

$$+ \frac{8\eta_{i-1}^2 L^2}{K}\mathbb{E}\|\tilde{d}_{i-1}\|^2 + \frac{256(I-1)L^2}{K\rho^2}\sum_{\ell=1}^{i-1}\eta_\ell^2\sum_{k=1}^K\mathbb{E}\|\nu_\ell^{(k)} - \bar{\nu}_\ell\|^2$$

The first inequality the assumption that $\alpha_i < 1$; the second inequality uses Lemma B.7 and Lemma B.8. This completes the proof. $\qquad\square$

**Lemma B.12.** *For $\tau \ge 0$ and $i \in [I]$. Suppose we have $\eta_{\tau,i} = \frac{\kappa}{(w_{\tau,i}+i+\tau I)^{1/3}}$, and have $\alpha_i < 1$, $w_{\tau,i} \ge 2$, $\eta_{\tau,i} \le \frac{\rho}{48LK^{0.5}I^2}$ be satisfied, we have:*

$$\frac{K\rho}{64L^2}\left(\frac{\mathbb{E}\|\bar{e}_{\tau+1}\|^2}{\eta_{\tau+1,I-1}} - \frac{\mathbb{E}\|\bar{e}_\tau\|^2}{\eta_{\tau,I-1}}\right) \le -\sum_{i=0}^{I-1}\frac{3\eta_{\tau+1,i}}{2\rho}\mathbb{E}\|\bar{e}_{\tau+1,i}\|^2 + \sum_{i=0}^{I-1}\frac{\eta_{\tau+1,i}\rho}{8}\mathbb{E}\|\tilde{d}_{\tau+1,i}\|^2$$

$$+ \sum_{i=0}^{I-1}\frac{\sigma^2 c^2\eta_{\tau+1,i}^3\rho}{16bL^2} + \frac{8I(I-1)}{\rho}\sum_{\ell=1}^I\eta_{\tau+1,\ell}\sum_{k=1}^K\mathbb{E}\|\nu_{\tau+1,\ell}^{(k)} - \bar{\nu}_{\tau+1,\ell}\|^2$$

*Proof.* Using Lemma B.11 at the global epoch $\tau - 1$, then for $i \geq 0$ (we denote $\eta_{\tau,-1} = \eta_{\tau-1,I-1}$ for all $\tau \geq 1$), we have:

$$\frac{\mathbb{E}\|\bar{e}_{\tau,i+1}\|^2}{\eta_{\tau,i}} - \frac{\mathbb{E}\|\bar{e}_{\tau,i}\|^2}{\eta_{\tau,i-1}}$$

$$\leq \left[\frac{(1-a_{\tau,i+1})^2}{\eta_{\tau,i}} - \frac{1}{\eta_{\tau,i-1}}\right]\mathbb{E}\|\bar{e}_{\tau,i}\|^2 + \frac{256(I-1)L^2}{\rho^2 K \eta_{\tau,i}}\sum_{\ell=1}^{i}\eta_{\tau,\ell}^2\sum_{k=1}^{K}\mathbb{E}\|\nu_{\tau,\ell}^{(k)} - \bar{\nu}_{\tau,\ell}\|^2$$

$$+ \frac{8L^2\eta_{\tau,i}}{K}\mathbb{E}\|\tilde{d}_{\tau,i}\|^2 + \frac{4a_{\tau,i+1}^2\sigma^2}{\eta_{\tau,i}bK}$$

$$\overset{(a)}{\leq} \left(\eta_{\tau,i}^{-1} - \eta_{\tau,i-1}^{-1} - c\eta_{\tau,i}\right)\mathbb{E}\|\bar{e}_{\tau,i}\|^2 + \frac{512(I-1)L^2}{\rho^2 K}\sum_{\ell=1}^{i}\eta_{\tau,\ell}\sum_{k=1}^{K}\mathbb{E}\|\nu_{\tau,\ell}^{(k)} - \bar{\nu}_{\tau,\ell}\|^2$$

$$+ \frac{8L^2\eta_{\tau,i}}{K}\mathbb{E}\|\tilde{d}_{\tau,i}\|^2 + \frac{4\sigma^2c^2\eta_{\tau,i}^3}{bK},$$

where inequality $(a)$ utilizes the fact that $(1-\alpha_{\tau,i})^2 \leq 1 - \alpha_{\tau,i} \leq 1$ and $a_{\tau,i+1} = c\eta_{\tau,i}^2$ for all $i \in [I]$, and the following fact: suppose we choose $\eta_{\tau,i} = \kappa/(w_i + i + \tau I)^{1/3}$, then for $0 \leq l \leq i < I$, we have:

$$\frac{\eta_{\tau,l}}{\eta_{\tau,i}} = \frac{(w_i + i + \tau I)^{1/3}}{(w_l + l + \tau I)^{1/3}} = \left(1 + \frac{w_i + i - w_l - l}{w_l + l + \tau I}\right)^{1/3}$$

$$\leq \left(1 + \frac{(I-1)}{w_l + l + \tau I}\right)^{1/3} \leq 1 + \frac{(I-1)}{3(w_l + l + \tau I)} \leq 2 \tag{17}$$

The first inequality is by the fact that $w_i \leq w_l$ and $0 < i - l < I - 1$, the second last inequality uses the concavity of $x^{1/3}$ as: $(x+y)^{1/3} - x^{1/3} \leq y/3x^{2/3}$, while the last inequality uses the fact that $w_l \geq 0, I \geq 1, l \geq 0, \tau \geq 1$.

For the difference $\eta_i^{-1} - \eta_{i-1}^{-1}$, we have:

$$\frac{1}{\eta_{\tau,i}} - \frac{1}{\eta_{\tau,i-1}} = \frac{(w_i + i + \tau I)^{1/3}}{\kappa} - \frac{(w_{i-1} + i - 1 + \tau I)^{1/3}}{\kappa}$$

$$\overset{(a)}{\leq} \frac{(w_{i-1} + i + \tau I)^{1/3}}{\kappa} - \frac{(w_{i-1} + i - 1 + \tau I)^{1/3}}{\kappa}$$

$$\overset{(b)}{\leq} \frac{1}{3\kappa(w_{i-1} + i - 1 + \tau I)^{2/3}} \overset{(c)}{\leq} \frac{2^{2/3}\kappa^2}{3\kappa^3(w_i + i + \tau I)^{2/3}} \overset{(e)}{=} \frac{2^{2/3}}{3\kappa^3}\eta_i^2 \overset{(d)}{\leq} \frac{\rho}{72\kappa^3 LK^{0.5}I^2}\eta_i, \tag{18}$$

where inequality $(a)$ is because that we choose $w \leq w$, $(b)$ results from the concavity of $x^{1/3}$ as: $(x+y)^{1/3} - x^{1/3} \leq y/(3x^{2/3})$, $(c)$ used the fact that $w \geq 2$, finally, $(d)$ and $(e)$ utilize the definition of $\eta_{\tau,i}$ and the condition that $\eta_{\tau,i} \leq \frac{\rho}{48LK^{0.5}I^2}$, respectively. So if we choose $c = \frac{96L^2}{K\rho^2} + \frac{\rho}{72\kappa^3 LK^{0.5}I^2}$ we have: $\eta_{\tau,i}^{-1} - \eta_{\tau,i-1}^{-1} - c\eta_{\tau,i} \leq -\frac{96L^2}{K\rho^2}\eta_{\tau,i}$,

Therefore, we have:

$$\frac{\mathbb{E}\|\bar{e}_{\tau,i+1}\|^2}{\eta_{\tau,i}} - \frac{\mathbb{E}\|\bar{e}_{\tau,i}\|^2}{\eta_{\tau,i-1}} \leq -\frac{96L^2\eta_{\tau,i}}{K\rho^2}\mathbb{E}\|\bar{e}_{\tau,i}\|^2 + \frac{512(I-1)L^2}{\rho^2 K}\sum_{\ell=1}^{i}\eta_{\tau,\ell}\sum_{k=1}^{K}\mathbb{E}\|\nu_{\tau,\ell}^{(k)} - \bar{\nu}_{\tau,\ell}\|^2$$

$$+ \frac{8L^2\eta_{\tau,i}}{K}\mathbb{E}\|\tilde{d}_{\tau,i}\|^2 + \frac{4\sigma^2c^2\eta_{\tau,i}^3}{bK},$$

Multiplying $K\rho/64L^2$ on both sides, we have:

$$\frac{K\rho}{64L^2}\left(\frac{\mathbb{E}\|\bar{e}_{\tau,i+1}\|^2}{\eta_{\tau,i}} - \frac{\mathbb{E}\|\bar{e}_{\tau,i}\|^2}{\eta_{\tau,i-1}}\right) \leq -\frac{3\eta_{\tau,i}}{2\rho}\mathbb{E}\|\bar{e}_{\tau,i}\|^2 + \frac{8(I-1)}{\rho}\sum_{\ell=1}^{i}\eta_{\tau,\ell}\sum_{k=1}^{K}\mathbb{E}\|\nu_{\tau,\ell}^{(k)} - \bar{\nu}_{\tau,\ell}\|^2$$

$$+ \frac{\eta_{\tau,i}\rho}{8}\mathbb{E}\|\tilde{d}_{\tau,i}\|^2 + \frac{\sigma^2c^2\eta_{\tau,i}^3\rho}{16L^2b}.$$

Then we sum the above inequality from 0 to $I-1$ and get:

$$
\begin{aligned}
\frac{K\rho}{64L^2}\left(\frac{\mathbb{E}\|\bar{e}_{\tau,I}\|^2}{\eta_{\tau,I-1}} - \frac{\mathbb{E}\|\bar{e}_{\tau,0}\|^2}{\eta_{\tau-1,I-1}}\right) &\leq -\sum_{i=0}^{I-1}\frac{3\eta_i}{2\rho}\mathbb{E}\|\bar{e}_{\tau,i}\|^2 + \sum_{i=0}^{I-1}\frac{8(I-1)}{\rho}\sum_{\ell=1}^{i}\eta_\ell\sum_{k=1}^{K}\mathbb{E}\|\nu_{\tau,\ell}^{(k)} - \bar{\nu}_{\tau,\ell}\|^2 \\
&\quad + \sum_{i=0}^{I-1}\frac{\eta_{\tau,i}\rho}{8}\mathbb{E}\|\tilde{d}_{\tau,i}\|^2 + \sum_{i=0}^{I-1}\frac{\sigma^2c^2\eta_{\tau,i}^3\rho}{16L^2b} \\
&\leq -\sum_{i=0}^{I-1}\frac{3\eta_i}{2\rho}\mathbb{E}\|\bar{e}_{\tau,i}\|^2 + \frac{8I(I-1)}{\rho}\sum_{\ell=1}^{I}\eta_\ell\sum_{k=1}^{K}\mathbb{E}\|\nu_{\tau,\ell}^{(k)} - \bar{\nu}_{\tau,\ell}\|^2 \\
&\quad + \sum_{i=0}^{I-1}\frac{\eta_{\tau,i}\rho}{8}\mathbb{E}\|\tilde{d}_{\tau,i}\|^2 + \sum_{i=0}^{I-1}\frac{\sigma^2c^2\eta_{\tau,i}^3\rho}{16L^2b}
\end{aligned}
$$

By definition, we have $\bar{e}_{\tau,0} = e_{\tau-1}$ and $\bar{e}_{\tau,I} = e_\tau$, then we get the results in the lemma by replacing $\tau$ by $\tau+1$. $\qquad\square$

### B.6 DESCENT IN POTENTIAL FUNCTION

We define the potential function as follows:

$$
\Phi_\tau := f(\tilde{x}_\tau) + \frac{K\rho}{64L^2}\frac{\|e_\tau\|^2}{\eta_{\tau-1,I-1}}. \tag{19}
$$

Next, we characterize the descent in the potential function.

**Lemma B.13.** *For any $\tau \geq 0$, we have:*

$$
\begin{aligned}
\mathbb{E}[\Phi_{\tau+1} - \Phi_\tau] &\leq -\sum_{i=0}^{I-1}\left(\frac{5\rho\eta_{\tau+1,i}}{8} - \frac{\eta_{\tau+1,i}^2 L}{2}\right)\mathbb{E}\|\tilde{d}_i\|^2 - \frac{1}{2\rho}\sum_{i=0}^{I-1}\eta_{\tau+1,i}\mathbb{E}\|\bar{e}_{\tau+1,i}\|^2 \\
&\quad + \frac{\sigma^2c^2\rho}{16L^2b}\sum_{i=0}^{I-1}\eta_{\tau+1,i}^3 + \frac{8I(I-1)}{\rho}\sum_{i=1}^{I}\eta_{\tau+1,i}\sum_{k=1}^{K}\mathbb{E}\|\nu_{\tau+1,i}^{(k)} - \bar{\nu}_{\tau+1,i}\|^2,
\end{aligned}
$$

*where the expectation is w.r.t the stochasticity of the algorithm.*

*Proof.* We can the inequality in the lemma by combining Lemma B.9 and Lemma B.12 $\qquad\square$

### B.7 ACCUMULATED GRADIENT ERROR

In this subsection, we bound the gradient consensus error given by term $\sum_{k=1}^{K}\mathbb{E}\|\nu_{\tau,i}^{(k)} - \bar{\nu}_{\tau,i}\|^2$.

**Lemma B.14.** *For $i \geq 1$ and $\alpha_i < 1$, we have:*

$$
\begin{aligned}
\sum_{k=1}^{K}\mathbb{E}\|\nu_{\tau,i}^{(k)} - \bar{\nu}_{\tau,i}\|^2 &\leq (1+\frac{1}{I})\sum_{k=1}^{K}\mathbb{E}\|\nu_{\tau,i-1}^{(k)} - \bar{\nu}_{\tau,i-1}\|^2 + 8KIL^2\eta_{\tau,i-1}^2\mathbb{E}\|\tilde{d}_{\tau,i-1}\|^2 + \frac{8KI\sigma^2c^2\eta_{\tau,i-1}^4}{b} \\
&\quad + 16KI\zeta^2c^2\eta_{\tau,i-1}^4 + \frac{96I^2L^2}{\rho^2}\sum_{\ell=1}^{i-1}\eta_{\tau,\ell^2}\sum_{k=1}^{K}\mathbb{E}\|\nu_{\tau,\ell}^{(k)} - \bar{\nu}_{\tau,\ell}\|^2
\end{aligned}
$$

*where the expectation is w.r.t. the stochasticity of the algorithm.*

*Proof.* By the update rule of $\nu_i^{(k)}$ (we omit the global epoch step for convenience), we have:

$$\mathbb{E}\|\nu_i^{(k)} - \bar{\nu}_i\|^2$$

$$= \mathbb{E}\left\|\nabla f^{(k)}(x_i^{(k)}; \mathcal{B}_i^{(k)}) + (1 - \alpha_i)\left(\nu_{i-1}^{(k)} - \nabla f^{(k)}(x_{i-1}^{(k)}; \mathcal{B}_i^{(k)})\right)\right.$$
$$\left. - \left(\frac{1}{K}\sum_{j=1}^{K}\nabla f^{(j)}(x_i^{(j)}; \mathcal{B}_i^{(j)}) + (1 - \alpha_i)\left(\bar{\nu}_{i-1} - \frac{1}{K}\sum_{j=1}^{K}\nabla f^{(j)}(x_{i-1}^{(j)}; \mathcal{B}_i^{(j)})\right)\right)\right\|^2$$

$$= \mathbb{E}\left\|(1 - \alpha_i)\left(\nu_{i-1}^{(k)} - \bar{\nu}_{i-1}\right) + \nabla f^{(k)}(x_i^{(k)}; \mathcal{B}_i^{(k)}) - \frac{1}{K}\sum_{j=1}^{K}\nabla f^{(j)}(x_i^{(j)}; \mathcal{B}_i^{(j)})\right.$$
$$\left. - (1 - \alpha_i)\left(\nabla f^{(k)}(x_{i-1}^{(k)}; \mathcal{B}_i^{(k)}) - \frac{1}{K}\sum_{j=1}^{K}\nabla f^{(j)}(x_{i-1}^{(j)}; \mathcal{B}_i^{(j)})\right)\right\|^2$$

$$\leq (1 + \beta)(1 - \alpha_i)^2 \mathbb{E}\left\|\nu_{i-1}^{(k)} - \bar{\nu}_{i-1}\right\|^2 + \left(1 + \frac{1}{\beta}\right)\mathbb{E}\left\|\nabla f^{(k)}(x_i^{(k)}; \mathcal{B}_i^{(k)}) - \frac{1}{K}\sum_{j=1}^{K}\nabla f^{(j)}(x_i^{(j)}; \mathcal{B}_i^{(j)})\right.$$
$$\left. - (1 - \alpha_i)\left(\nabla f^{(k)}(x_{i-1}^{(k)}; \mathcal{B}_i^{(k)}) - \frac{1}{K}\sum_{j=1}^{K}\nabla f^{(j)}(x_{i-1}^{(j)}; \mathcal{B}_i^{(j)})\right)\right\|^2, \tag{20}$$

where the last inequality uses Proposition B.1.

Next, we consider the second term:

$$\mathbb{E}\left\|\nabla f^{(k)}(x_i^{(k)}; \mathcal{B}_i^{(k)}) - \frac{1}{K}\sum_{j=1}^{K}\nabla f^{(j)}(x_i^{(j)}; \mathcal{B}_i^{(j)})\right.$$
$$\left. - (1 - \alpha_i)\left(\nabla f^{(k)}(x_{i-1}^{(k)}; \mathcal{B}_i^{(k)}) - \frac{1}{K}\sum_{j=1}^{K}\nabla f^{(j)}(x_{i-1}^{(j)}; \mathcal{B}_i^{(j)})\right)\right\|^2$$

$$\overset{(a)}{\leq} 2\mathbb{E}\left\|\nabla f^{(k)}(x_i^{(k)}; \mathcal{B}_i^{(k)}) - \frac{1}{K}\sum_{j=1}^{K}\nabla f^{(j)}(x_i^{(j)}; \mathcal{B}_i^{(j)})\right.$$
$$\left. - \left(\nabla f^{(k)}(x_{i-1}^{(k)}; \mathcal{B}_i^{(k)}) - \frac{1}{K}\sum_{j=1}^{K}\nabla f^{(j)}(x_{i-1}^{(j)}; \mathcal{B}_i^{(j)})\right)\right\|^2$$
$$+ 2\alpha_i^2 \mathbb{E}\left\|\nabla f^{(k)}(x_{i-1}^{(k)}; \mathcal{B}_i^{(k)}) - \frac{1}{K}\sum_{j=1}^{K}\nabla f^{(j)}(x_{i-1}^{(j)}; \mathcal{B}_i^{(j)})\right\|^2$$

$$\overset{(b)}{\leq} 2\mathbb{E}\left\|\left(\nabla f^{(k)}(x_i^{(k)}; \mathcal{B}_i^{(k)}) - \nabla f^{(k)}(x_{i-1}^{(k)}; \mathcal{B}_i^{(k)})\right)\right\|^2$$
$$+ 2\alpha_i^2 \mathbb{E}\left\|\nabla f^{(k)}(x_{i-1}^{(k)}; \mathcal{B}_i^{(k)}) - \frac{1}{K}\sum_{j=1}^{K}\nabla f^{(j)}(x_{i-1}^{(j)}; \mathcal{B}_i^{(j)})\right\|^2$$

$$\overset{(c)}{\leq} 2L^2 \mathbb{E}\left\|x_i^{(k)} - x_{i-1}^{(k)}\right\|^2 + 2\alpha_i^2 \mathbb{E}\left\|\nabla f^{(k)}(x_{i-1}^{(k)}; \mathcal{B}_i^{(k)}) - \frac{1}{K}\sum_{j=1}^{K}\nabla f^{(j)}(x_{i-1}^{(j)}; \mathcal{B}_i^{(j)})\right\|^2, \tag{21}$$

where inequality (a) uses Proposition B.1; inequality (b) uses Proposition B.2; inequality (c) uses the smoothness assumption.

Next, we consider the second term in equation 21 above, we have

$$
\mathbb{E}\left\|\nabla f^{(k)}(x_{i-1}^{(k)}; \mathcal{B}_i^{(k)}) - \frac{1}{K}\sum_{j=1}^{K}\nabla f^{(j)}(x_{i-1}^{(j)}; \mathcal{B}_i^{(j)})\right\|^2
$$

$$
= \mathbb{E}\left\|\left(\nabla f^{(k)}(x_{i-1}^{(k)}; \mathcal{B}_i^{(k)}) - \nabla f^{(k)}(x_{i-1}^{(k)})\right)\right.
$$
$$
\left. - \frac{1}{K}\sum_{j=1}^{K}\left(\nabla f^{(j)}(x_{i-1}^{(j)}; \mathcal{B}_i^{(j)}) - \nabla f^{(j)}(x_{i-1}^{(j)})\right) + \nabla f^{(k)}(x_{i-1}^{(k)}) - \frac{1}{K}\sum_{j=1}^{K}\nabla f^{(j)}(x_{i-1}^{(j)})\right\|^2
$$

$$
\leq 2\mathbb{E}\left\|\left(\nabla f^{(k)}(x_{i-1}^{(k)}; \mathcal{B}_i^{(k)}) - \nabla f^{(k)}(x_{i-1}^{(k)})\right)\right.
$$
$$
\left. - \frac{1}{K}\sum_{j=1}^{K}\left(\nabla f^{(j)}(x_{i-1}^{(j)}; \mathcal{B}_i^{(j)}) - \nabla f^{(j)}(x_{i-1}^{(j)})\right)\right\|^2
$$
$$
+ 2\mathbb{E}\left\|\nabla f^{(k)}(x_{i-1}^{(k)}) - \frac{1}{K}\sum_{j=1}^{K}\nabla f^{(j)}(x_{i-1}^{(j)})\right\|^2
$$

$$
\overset{(a)}{\leq} 2\mathbb{E}\left\|\left(\nabla f^{(k)}(x_{i-1}^{(k)}; \mathcal{B}_i^{(k)}) - \nabla f^{(k)}(x_{i-1}^{(k)})\right)\right\|^2 + 2\mathbb{E}\left\|\nabla f^{(k)}(x_{i-1}^{(k)}) - \frac{1}{K}\sum_{j=1}^{K}\nabla f^{(j)}(x_{i-1}^{(j)})\right\|^2
$$

$$
\leq 2\mathbb{E}\left\|\left(\nabla f^{(k)}(x_{i-1}^{(k)}; \mathcal{B}_i^{(k)}) - \nabla f^{(k)}(x_{i-1}^{(k)})\right)\right\|^2 + 4\mathbb{E}\left\|\nabla f^{(k)}(\tilde{x}_{i-1}) - \nabla f(\tilde{x}_{i-1})\right\|^2
$$
$$
+ 8\mathbb{E}\left\|\nabla f^{(k)}(x_{i-1}^{(k)}) - \nabla f^{(k)}(\tilde{x}_{i-1})\right\|^2 + 8\mathbb{E}\left\|\nabla f(\tilde{x}_{i-1}) - \frac{1}{K}\sum_{j=1}^{K}\nabla f^{(j)}(x_{i-1}^{(j)})\right\|^2
$$

$$
\overset{(b)}{\leq} \frac{2\sigma^2}{b} + \frac{4}{K}\sum_{j=1}^{K}\mathbb{E}\|\nabla f^{(k)}(\tilde{x}_{i-1}) - \nabla f^{(j)}(\bar{x}_{i-1})\|^2
$$
$$
+ 8L^2\mathbb{E}\|x_{i-1}^{(k)} - \tilde{x}_{i-1}\|^2 + \frac{8L^2}{K}\sum_{j=1}^{K}\mathbb{E}\|x_{i-1}^{(j)} - \tilde{x}_{i-1}\|^2
$$

$$
\overset{(c)}{\leq} \frac{2\sigma^2}{b} + 4\zeta^2 + 8L^2\mathbb{E}\|x_{i-1}^{(k)} - \tilde{x}_{i-1}\|^2 + \frac{8L^2}{K}\sum_{j=1}^{K}\mathbb{E}\|x_{i-1}^{(j)} - \tilde{x}_{i-1}\|^2, \tag{22}
$$

where inequality $(a)$ uses Proposition B.2; inequality $(b)$ utilizes bounded variance assumption; $(c)$ uses the bounded heterogeneity assumption. Finally, substituting equation 22 and equation 21 into equation 20 and sum over all K workers, we get

$$
\sum_{k=1}^{K}\mathbb{E}\|\nu_i^{(k)} - \bar{\nu}_i\|^2
$$

$$
\leq (1-\alpha_i)^2(1+\beta)\sum_{k=1}^{K}\mathbb{E}\|\nu_{i-1}^{(k)} - \bar{\nu}_{i-1}\|^2 + 2L^2\left(1+\frac{1}{\beta}\right)\sum_{k=1}^{K}\mathbb{E}\|x_i^{(k)} - x_{i-1}^{(k)}\|^2
$$
$$
+ \frac{4K\sigma^2}{b}\left(1+\frac{1}{\beta}\right)\alpha_i^2 + 8K\zeta^2\left(1+\frac{1}{\beta}\right)\alpha_i^2 + 32L^2\left(1+\frac{1}{\beta}\right)\alpha_i^2\sum_{k=1}^{K}\mathbb{E}\|x_{i-1}^{(k)} - \tilde{x}_{i-1}\|^2
$$

where the second inequality uses claim 3 of the Lemma B.6. For the term $\sum_{k=1}^{K}\mathbb{E}\|x_i^{(k)} - x_{i-1}^{(k)}\|^2$, we have:

$$
\sum_{k=1}^{K}\mathbb{E}\|x_i^{(k)} - x_{i-1}^{(k)}\|^2 \leq 2\sum_{k=1}^{K}\mathbb{E}\|x_i^{(k)} - \tilde{x}_i - (x_{i-1}^{(k)} - \tilde{x}_{i-1})\|^2 + 2\sum_{k=1}^{K}\mathbb{E}\|\tilde{x}_i - \tilde{x}_{i-1}\|^2
$$

$$
\leq 4\sum_{k=1}^{K}\mathbb{E}\|x_i^{(k)} - \tilde{x}_i\|^2 + 4\sum_{k=1}^{K}\mathbb{E}\|x_{i-1}^{(k)} - \tilde{x}_{i-1}\|^2 + 2K\eta_{i-1}^2\mathbb{E}\|\tilde{d}_{i-1}\|^2
$$

So we have:

$$\sum_{k=1}^{K} \mathbb{E}\|\nu_i^{(k)} - \bar{\nu}_i\|^2$$

$$\leq (1-\alpha_i)^2(1+\beta) \sum_{k=1}^{K} \mathbb{E}\big\|\nu_{i-1}^{(k)} - \bar{\nu}_{i-1}\big\|^2 + 4KL^2\eta_{i-1}^2\Big(1+\frac{1}{\beta}\Big)\mathbb{E}\|\tilde{d}_{i-1}\|^2$$

$$+ 8L^2\Big(1+\frac{1}{\beta}\Big) \sum_{k=1}^{K} \mathbb{E}\|x_i^{(k)} - \tilde{x}_i\|^2$$

$$+ \frac{4K\sigma^2}{b}\Big(1+\frac{1}{\beta}\Big)\alpha_i^2 + 8K\zeta^2\Big(1+\frac{1}{\beta}\Big)\alpha_i^2 + 40L^2\Big(1+\frac{1}{\beta}\Big) \sum_{k=1}^{K} \mathbb{E}\|x_{i-1}^{(k)} - \tilde{x}_{i-1}\|^2$$

Next using Lemma B.7, we have:

$$\sum_{k=1}^{K} \mathbb{E}\|\nu_i^{(k)} - \bar{\nu}_i\|^2$$

$$\leq (1-\alpha_i)^2(1+\beta) \sum_{k=1}^{K} \mathbb{E}\big\|\nu_{i-1}^{(k)} - \bar{\nu}_{i-1}\big\|^2 + 4KL^2\eta_{i-1}^2\Big(1+\frac{1}{\beta}\Big)\mathbb{E}\|\tilde{d}_{i-1}\|^2$$

$$+ \frac{8L^2}{\rho^2}\Big(1+\frac{1}{\beta}\Big) \sum_{k=1}^{K} \mathbb{E}\|z_i^{(k)} - \bar{z}_i\|^2$$

$$+ \frac{4K\sigma^2}{b}\Big(1+\frac{1}{\beta}\Big)\alpha_i^2 + 8K\zeta^2\Big(1+\frac{1}{\beta}\Big)\alpha_i^2 + \frac{40L^2}{\rho^2}\Big(1+\frac{1}{\beta}\Big) \sum_{k=1}^{K} \mathbb{E}\|z_{i-1}^{(k)} - \bar{z}_{i-1}\|^2$$

$$\leq (1-\alpha_i)^2(1+\beta) \sum_{k=1}^{K} \mathbb{E}\big\|\nu_{i-1}^{(k)} - \bar{\nu}_{i-1}\big\|^2 + 4KL^2\eta_{i-1}^2\Big(1+\frac{1}{\beta}\Big)\mathbb{E}\|\tilde{d}_{i-1}\|^2$$

$$+ \frac{4K\sigma^2}{b}\Big(1+\frac{1}{\beta}\Big)\alpha_i^2 + 8K\zeta^2\Big(1+\frac{1}{\beta}\Big)\alpha_i^2$$

$$+ \frac{48L^2}{\rho^2}\Big(1+\frac{1}{\beta}\Big)(I-1) \sum_{\ell=1}^{i-1}\eta_\ell^2 \sum_{k=1}^{K} \mathbb{E}\|\nu_\ell^{(k)} - \bar{\nu}_\ell\|^2$$

$$\leq (1+\frac{1}{I}) \sum_{k=1}^{K} \mathbb{E}\big\|\nu_{i-1}^{(k)} - \bar{\nu}_{i-1}\big\|^2 + 8KIL^2\eta_{i-1}^2\mathbb{E}\|\tilde{d}_{i-1}\|^2 + \frac{8KI\sigma^2c^2\eta_{i-1}^4}{b}$$

$$+ 16KI\zeta^2c^2\eta_{i-1}^4 + \frac{96I^2L^2}{\rho^2}\sum_{\ell=1}^{i-1}\eta_\ell^2 \sum_{k=1}^{K} \mathbb{E}\|\nu_\ell^{(k)} - \bar{\nu}_\ell\|^2 \tag{23}$$

In the last inequality, we choose $\beta = 1/I$, then we have $(1+1/\beta) \leq (1+I) \leq 2I$, we also use the fact that $(1-\alpha_i)^2 < 1$ and $a_i = c\eta_{i-1}^2 < 1$. This completes the proof. $\qquad\square$

**Lemma B.15.** *For $\eta_i \leq \frac{\rho}{48LK^{0.5}I^2}$, then we have*

$$\frac{I^2}{\rho}\sum_{i=1}^{I}\eta_i\sum_{k=1}^{K}\mathbb{E}\|\nu_i^{(k)} - \bar{\nu}_i\|^2 \leq \frac{\rho}{84}\sum_{i=0}^{I-1}\eta_i\mathbb{E}\|\tilde{d}_i\|^2 + \left(\frac{\rho\sigma^2 c^2}{84bL^2} + \frac{\rho\zeta^2 c^2}{42L^2}\right)\sum_{i=0}^{I-1}\eta_i^3$$

*Proof.* By Lemma B.14 (we omit the global epoch number for convenience) we have:

$$\sum_{k=1}^{K}\mathbb{E}\|\nu_i^{(k)} - \bar{\nu}_i\|^2 \leq \left(1 + \frac{1}{I}\right)\sum_{k=1}^{K}\mathbb{E}\|\nu_{i-1}^{(k)} - \bar{\nu}_{i-1}\|^2 + 8KIL^2\eta_{i-1}^2\mathbb{E}\|\tilde{d}_{i-1}\|^2 + \frac{8KI\sigma^2 c^2\eta_{i-1}^4}{b}$$

$$+ 16KI\zeta^2 c^2\eta_{i-1}^4 + \frac{96I^2L^2}{\rho^2}\sum_{\ell=1}^{i-1}\eta_\ell^2\sum_{k=1}^{K}\mathbb{E}\|\nu_\ell^{(k)} - \bar{\nu}_\ell\|^2$$

$$\leq \left(1 + \frac{1}{I}\right)\sum_{k=1}^{K}\mathbb{E}\|\nu_{i-1}^{(k)} - \bar{\nu}_{i-1}\|^2 + \frac{\sqrt{K}L\rho\eta_{i-1}}{6I}\mathbb{E}\|\tilde{d}_{i-1}\|^2 + \frac{\sqrt{K}\rho\sigma^2 c^2\eta_{i-1}^3}{6ILb}$$

$$+ \frac{\sqrt{K}\rho\zeta^2 c^2\eta_{i-1}^3}{3IL} + \frac{96I^2L^2}{\rho^2}\sum_{\ell=1}^{i-1}\eta_\ell^2\sum_{k=1}^{K}\mathbb{E}\|\nu_\ell^{(k)} - \bar{\nu}_\ell\|^2, \tag{24}$$

where in the second inequality, we use the condition that $\eta_i \leq \frac{\rho}{48LK^{0.5}I^2}$. Applying equation 24 recursively from 1 to $i$. We have:

$$\sum_{k=1}^{K}\mathbb{E}\|\nu_i^{(k)} - \bar{\nu}_i\|^2 \leq \frac{\sqrt{K}L\rho}{6I}\sum_{\ell=0}^{i-1}\left(1 + \frac{1}{I}\right)^{i-1-\ell}\eta_\ell\mathbb{E}\|\tilde{d}_\ell\|^2 + \frac{\sqrt{K}\rho\sigma^2 c^2}{6ILb}\sum_{\ell=0}^{i-1}\left(1 + \frac{1}{I}\right)^{i-1-\ell}\eta_\ell^3$$

$$+ \frac{\sqrt{K}\rho\zeta^2 c^2}{3IL}\sum_{\ell=0}^{i-1}\left(1 + \frac{1}{I}\right)^{i-1-\ell}\eta_\ell^3$$

$$+ \frac{96L^2I^2}{\rho^2}\sum_{\ell=0}^{i-1}\left(1 + \frac{1}{I}\right)^{i-1-\ell}\sum_{\bar{\ell}=0}^{\ell}\eta_{\bar{\ell}}^2\sum_{k=1}^{K}\mathbb{E}\|\nu_{\bar{\ell}}^{(k)} - \bar{\nu}_{\bar{\ell}}\|^2$$

$$\overset{(a)}{\leq} \frac{\sqrt{K}L\rho}{6I}\left(1 + \frac{1}{I}\right)^{I}\sum_{\ell=0}^{i-1}\eta_\ell\mathbb{E}\|\tilde{d}_\ell\|^2 + \frac{\sqrt{K}\rho\sigma^2 c^2}{6ILb}\left(1 + \frac{1}{I}\right)^{I}\sum_{\ell=0}^{i-1}\eta_\ell^3$$

$$+ \frac{\sqrt{K}\rho\zeta^2 c^2}{3IL}\left(1 + \frac{1}{I}\right)^{I}\sum_{\ell=0}^{i-1}\eta_\ell^3 + \frac{96L^2I^3}{\rho^2}\left(1 + \frac{1}{I}\right)^{I}\sum_{\bar{\ell}=0}^{i-1}\eta_{\bar{\ell}}^2\sum_{k=1}^{K}\mathbb{E}\|\nu_{\bar{\ell}}^{(k)} - \bar{\nu}_{\bar{\ell}}\|^2$$

$$\overset{(b)}{\leq} \frac{\sqrt{K}L\rho}{2I}\sum_{\ell=0}^{i-1}\eta_\ell\mathbb{E}\|\tilde{d}_\ell\|^2 + \frac{\sqrt{K}\rho\sigma^2 c^2}{2ILb}\sum_{\ell=0}^{i-1}\eta_\ell^3 + \frac{\sqrt{K}\rho\zeta^2 c^2}{IL}\sum_{\ell=0}^{i-1}\eta_\ell^3$$

$$+ \frac{288L^2I^3}{\rho^2}\sum_{\ell=0}^{i-1}\eta_\ell^2\sum_{k=1}^{K}\mathbb{E}\|\nu_\ell^{(k)} - \bar{\nu}_\ell\|^2, \tag{25}$$

where inequality $(a)$ is by the fact that $1 + 1/I > 1$ and $i - 1 - \ell \leq I$ for $i \in [I]$ and $\ell \in [i]$ and inequality $(b)$ is because that $(1 + 1/I)^I \leq e < 3$.

Next, multiplying both sides of equation 25 by $\eta_i$ and summing over $i = 1$ to $I$:

$$
\sum_{i=1}^{I} \eta_i \sum_{k=1}^{K} \mathbb{E}\|\nu_i^{(k)} - \bar{\nu}_i\|^2 \leq \frac{\sqrt{K}L\rho}{2I} \sum_{i=1}^{I} \eta_i \sum_{\ell=0}^{i-1} \eta_\ell \mathbb{E}\|\tilde{d}_\ell\|^2 + \frac{\sqrt{K}\rho\sigma^2 c^2}{2ILb} \sum_{i=1}^{I} \eta_i \sum_{\ell=0}^{i-1} \eta_\ell^3
$$

$$
+ \frac{\sqrt{K}\rho\zeta^2 c^2}{IL} \sum_{i=1}^{I} \eta_i \sum_{\ell=0}^{i-1} \eta_\ell^3 + \frac{288L^2I^3}{\rho^2} \sum_{i=1}^{I} \eta_i \sum_{\ell=0}^{i-1} \eta_\ell^2 \sum_{k=1}^{K} \mathbb{E}\|\nu_\ell^{(k)} - \bar{\nu}_\ell\|^2
$$

$$
\overset{(a)}{\leq} \frac{\sqrt{K}L\rho}{2I} \left(\sum_{i=1}^{I} \eta_i\right) \sum_{\ell=0}^{I-1} \eta_\ell \mathbb{E}\|\tilde{d}_\ell\|^2 + \left(\frac{\sqrt{K}\rho\sigma^2 c^2}{2ILb} + \frac{\sqrt{K}\rho\zeta^2 c^2}{IL}\right) \left(\sum_{i=1}^{I} \eta_i\right) \sum_{\ell=0}^{I-1} \eta_\ell^3
$$

$$
+ \frac{288L^2I^3}{\rho^2} \left(\sum_{i=1}^{I} \eta_i\right) \sum_{\ell=0}^{I-1} \eta_\ell^2 \sum_{k=1}^{K} \mathbb{E}\|\nu_\ell^{(k)} - \bar{\nu}_\ell\|^2
$$

$$
\overset{(b)}{\leq} \frac{\rho^2}{96I^2} \sum_{i=0}^{I-1} \eta_i \mathbb{E}\|\tilde{d}_i\|^2 + \left(\frac{\rho^2\sigma^2 c^2}{96I^2L^2b} + \frac{\rho^2\zeta^2 c^2}{48I^2L^2}\right) \sum_{i=0}^{I-1} \eta_i^3 + \frac{1}{8} \sum_{\ell=1}^{I-1} \eta_\ell \sum_{k=1}^{K} \mathbb{E}\|\nu_\ell^{(k)} - \bar{\nu}_\ell\|^2
$$

where inequality $(a)$ uses the fact that $i \leq I$ and $(b)$ uses that we choose $\eta_i \leq \rho/(48LK^{0.5}I^2)$. Rearranging the terms we have:

$$
\frac{7}{8} \sum_{i=1}^{I} \eta_i \sum_{k=1}^{K} \mathbb{E}\|\nu_i^{(k)} - \bar{\nu}_i\|^2 \leq \frac{\rho^2}{96I^2} \sum_{i=0}^{I-1} \eta_i \mathbb{E}\|\tilde{d}_i\|^2 + \left(\frac{\rho^2\sigma^2 c^2}{96I^2L^2b} + \frac{\rho^2\zeta^2 c^2}{48I^2L^2}\right) \sum_{i=0}^{I-1} \eta_i^3
$$

Multiplying $8I^2/(7K\rho)$ on both sides, we have:

$$
\frac{I^2}{\rho} \sum_{i=1}^{I} \eta_i \sum_{k=1}^{K} \mathbb{E}\|\nu_i^{(k)} - \bar{\nu}_i\|^2 \leq \frac{\rho}{84} \sum_{i=0}^{I-1} \eta_i \mathbb{E}\|\tilde{d}_i\|^2 + \left(\frac{\rho\sigma^2 c^2}{84L^2b} + \frac{\rho\zeta^2 c^2}{42L^2}\right) \sum_{i=0}^{I-1} \eta_i^3
$$

This completes the proof. $\qquad\square$

### B.8 PROOF OF THE MAIN CONVERGENCE THEOREM

In this subsection, we prove Theorem 5.6 and Corollary 5.7. To prove Theorem 5.6, we firstly show the following theorem hold:

**Theorem B.16.** *Choosing the parameters as* $\kappa = \dfrac{K^{2/3}\rho}{L}$, $c = \dfrac{96L^2}{K\rho^2} + \dfrac{\rho}{72\kappa^3 LK^{0.5}I^2}$, $w_t = \max\{2, 48^3I^6K^{7/2} - t\}$, *and choose* $\eta_t = \dfrac{\kappa}{(w_t+t)^{1/3}}$, *then we have:*

$$
\frac{1}{T} \sum_{t=0}^{T-1} \left(\mathbb{E}\|\tilde{d}_t\|^2 + \frac{1}{\rho^2}\mathbb{E}\|\bar{e}_t\|^2\right)
$$

$$
\leq \left[\frac{96LI^2}{\rho^2 T} + \frac{2L}{\rho^2 K^{2/3}T^{2/3}}\right](f(x_0) - f^*) + \left[\frac{72I^4}{b_1\rho^2 T} + \frac{3I^2}{2b_1\rho^2 K^{2/3}T^{2/3}}\right]\sigma^2
$$

$$
+ \frac{192^2}{\rho^2} \times \left(\frac{48I^2}{T} + \frac{1}{K^{2/3}T^{2/3}}\right) \times \left(\frac{\sigma^2}{4b} + \frac{2\zeta^2}{21}\right)\log(T+1).
$$

*Proof.* By definition, we have $\eta_t \leq \eta_0 < \kappa/w_0^{1/3} = \rho/48LK^{0.5}I^2$, then $c = \dfrac{L^2}{K\rho^2}\left(96 + \dfrac{1}{72K^{1.5}I^2}\right) \leq \dfrac{192L^2}{K\rho^2}$ and: $c\eta_t^2 \leq c\eta_0^2 = \dfrac{192L^2}{K\rho^2} * \dfrac{\rho^2}{48^2L^2KI^4} < 1$, so we have $\alpha_t < 1$, then the conditions of Lemma B.13-Lemma B.15 are satisfied.

Firstly, substitute the gradient consensus error in Lemma B.15 to Lemma B.13, we can write the descent of potential function as:

$$\mathbb{E}[\Phi_{\tau+1} - \Phi_\tau] \leq -\sum_{i=0}^{I-1} \left( \frac{5\rho\eta_{\tau+1,i}}{8} - \frac{\eta_{\tau+1,i}^2 L}{2} \right) \mathbb{E}\|\tilde{d}_{\tau+1,i}\|^2 - \frac{1}{2\rho} \sum_{i=0}^{I-1} \eta_{\tau+1,i} \mathbb{E}\|\bar{e}_{\tau+1,i}\|^2$$

$$+ \frac{\sigma^2 c^2 \rho}{16L^2 b} \sum_{i=0}^{I-1} \eta_{\tau+1,i}^3 + \frac{2\rho}{21} \sum_{i=0}^{I-1} \eta_{\tau+1,i} \mathbb{E}\|\tilde{d}_{\tau+1,i}\|^2 + \left( \frac{2\rho\sigma^2 c^2}{21L^2 b} + \frac{4\rho\zeta^2 c^2}{21L^2} \right) \sum_{i=0}^{I-1} \eta_{\tau+1,i}^3$$

$$\overset{(a)}{\leq} -\sum_{i=0}^{I-1} \frac{\rho\eta_i}{2} \mathbb{E}\|\tilde{d}_{\tau+1,i}\|^2 - \frac{1}{2\rho} \sum_{i=0}^{I-1} \eta_{\tau+1,i} \mathbb{E}\|\bar{e}_{\tau+1,i}\|^2 + \left( \frac{\rho\sigma^2 c^2}{4L^2 b} + \frac{\rho\zeta^2 c^2}{4L^2} \right) \sum_{i=0}^{I-1} \eta_{\tau+1,i}^3,$$

where $(a)$ follows from the fact that $\eta_i \leq \frac{\rho}{48LK^{0.5}I^2} \leq \frac{\rho}{48L}$.

Suppose we denote $T = EI$, and $t = \tau I + i$ for $t \geq 0$ and $\tau \geq 0$. Then we have $\eta_t = \eta_{\tau+1,i}$, $\tilde{d}_t = \tilde{d}_{\tau+1,i}$, $\bar{e}_t = \bar{e}_{\tau+1,i}$. In particular, we denote $\eta_{-1} = \eta_0$ for convenience.

Then we sum the above inequality for $\tau$ from 0 to $E-1$, and get:

$$\mathbb{E}[\Phi_E - \Phi_0] \leq -\sum_{t=0}^{T-1} \left( \frac{\rho\eta_t}{2} \right) \mathbb{E}\|\tilde{d}_t\|^2 - \sum_{t=0}^{T-1} \frac{\eta_t}{2\rho} \mathbb{E}\|\bar{e}_t\|^2 + \left( \frac{\rho\sigma^2 c^2}{4L^2 b} + \frac{\rho\zeta^2 c^2}{4L^2} \right) \sum_{t=0}^{T} \eta_t^3,$$

Rearranging terms, we get:

$$\sum_{t=1}^{T} \left( \frac{\rho\eta_t}{2} \mathbb{E}\|\tilde{d}_t\|^2 + \frac{\eta_t}{2\rho} \mathbb{E}\|\bar{e}_t\|^2 \right) \leq \mathbb{E}[\Phi_0 - \Phi_E] + \left( \frac{\rho\sigma^2 c^2}{4L^2 b} + \frac{\rho\zeta^2 c^2}{4L^2} \right) \sum_{t=0}^{T-1} \eta_t^3$$

$$\overset{(a)}{\leq} f(x_0) - f^* + \frac{K\rho}{64L^2} \frac{\mathbb{E}\|e_0\|^2}{\eta_0} + \left( \frac{\rho\sigma^2 c^2}{4L^2 b} + \frac{\rho\zeta^2 c^2}{4L^2} \right) \sum_{t=0}^{T-1} \eta_t^3$$

$$\overset{(b)}{\leq} f(x_0) - f^* + \frac{\sigma^2 \rho}{64b_1 L^2 \eta_0} + \left( \frac{\rho\sigma^2 c^2}{4L^2 b} + \frac{\rho\zeta^2 c^2}{4L^2} \right) \sum_{t=0}^{T-1} \eta_t^3, \quad (26)$$

where $(a)$ follows from the fact that $f^* \leq \Phi_E$ and $(b)$ results from application of Lemma B.10 and $b$ is the minibatch size at the first iteration.

Next for the last term of the equation 26 above, we have:

$$\sum_{t=0}^{T-1} \eta_t^3 = \sum_{t=0}^{T-1} \frac{\kappa^3}{w_t + t} \overset{(a)}{\leq} \sum_{t=0}^{T-1} \frac{\kappa^3}{1 + t} \overset{(b)}{\leq} \kappa^3 \ln(T+1). \quad (27)$$

where inequality $(a)$ above follows from the fact that we have $w_t > 1$ and inequality $(b)$ follows from the application of Proposition B.3.

Substituting equation 27 in equation 26, multiplying both sides by $2/(\rho\eta_T T)$ and using the fact that $\eta_t$ is non-increasing in $t$ we have

$$\frac{1}{T} \sum_{t=0}^{T-1} \left( \mathbb{E}\|\tilde{d}_t\|^2 + \frac{1}{\rho^2} \mathbb{E}\|\bar{e}_t\|^2 \right) \leq \frac{2(f(x_0) - f^*)}{\rho\eta_T T} + \frac{1}{\eta_T T} \frac{\sigma^2}{32b_1 L^2 \eta_0} + \frac{\kappa^3}{\eta_T T} \left( \frac{\sigma^2 c^2}{4bL^2} + \frac{2\zeta^2 c^2}{21L^2} \right) \ln(T+1).$$

$$(28)$$

Now considering each term of equation 28 above separately. For the first term:

$$\frac{1}{\eta_T T} = \frac{(w_T + T)^{1/3}}{\kappa T} \overset{(a)}{\leq} \frac{w_T^{1/3}}{\kappa T} + \frac{1}{\kappa T^{2/3}} \overset{(b)}{\leq} \frac{48LI^2 K^{0.5}}{\rho T} + \frac{L}{\rho K^{2/3} T^{2/3}}. \quad (29)$$

where inequality $(a)$ follows from identity $(x + y)^{1/3} \leq x^{1/3} + y^{1/3}$ and inequality $(b)$ follows from the definition of $\kappa$ and $w_T$ Similarly, for the second term of equation 28, we have from the definition

of $\eta_0$ and $\eta_T$:

$$\frac{1}{\eta_T T} \frac{\sigma^2}{32bL^2\eta_0} \leq \left(\frac{48LK^{0.5}I^2}{\rho T} + \frac{L}{\rho K^{2/3}T^{2/3}}\right) \times \frac{\sigma^2}{32b_1 L^2} \times \frac{w_0^{1/3}}{\kappa}$$

$$\leq \left(\frac{48LK^{0.5}I^2}{\rho T} + \frac{L}{\rho K^{2/3}T^{2/3}}\right) \times \frac{\sigma^2}{32b_1 L^2} \times \frac{48LK^{0.5}I^2}{\rho}$$

$$\leq \frac{72KI^4}{b_1\rho^2 T}\sigma^2 + \frac{3K^{0.5}I^2}{2b_1\rho^2 K^{2/3}T^{2/3}}\sigma^2. \tag{30}$$

Finally, for the last term in equation 28 above, we have from the definition of the stepsize, $\eta_t$,

$$\frac{\kappa^3 c^2}{\eta_T T L^2}\left(\frac{\sigma^2}{4b} + \frac{2\zeta^2}{21}\right)\ln(T+1)$$

$$\leq \left(\frac{48LK^{0.5}I^2}{\rho T} + \frac{L}{\rho K^{2/3}T^{2/3}}\right) \times \frac{192^2}{L\rho} \times \left(\frac{\sigma^2}{4b} + \frac{2\zeta^2}{21}\right)\log(T+1)$$

$$\leq \frac{192^2}{\rho^2} \times \left(\frac{48K^{0.5}I^2}{T} + \frac{1}{K^{2/3}T^{2/3}}\right) \times \left(\frac{\sigma^2}{4b} + \frac{2\zeta^2}{21}\right)\log(T+1). \tag{31}$$

Finally, substituting the bounds obtained in equation 29, equation 30 and equation 31 into equation 28, we get

$$\frac{1}{T}\sum_{t=0}^{T-1}\left(\mathbb{E}\|\tilde{d}_t\|^2 + \frac{1}{\rho^2}\mathbb{E}\|\bar{e}_t\|^2\right)$$

$$\leq \left[\frac{96LK^{0.5}I^2}{\rho^2 T} + \frac{2L}{\rho^2 K^{2/3}T^{2/3}}\right](f(x_0) - f^*) + \left[\frac{72KI^4}{b_1\rho^2 T} + \frac{3K^{0.5}I^2}{2b_1\rho^2 K^{2/3}T^{2/3}}\right]\sigma^2$$

$$+ \frac{192^2}{\rho^2} \times \left(\frac{48K^{0.5}I^2}{T} + \frac{1}{K^{2/3}T^{2/3}}\right) \times \left(\frac{\sigma^2}{4b} + \frac{2\zeta^2}{21}\right)\log(T+1).$$

This completes the proof of the theorem. $\qquad\qquad\square$

Now we are ready to show Theorem 5.6. Firstly notice that:

$$\frac{\mathcal{G}_t}{\rho^2} = \frac{1}{\eta_t^2}\|\tilde{x}_t - \tilde{x}_{t+1}\|^2 + \frac{1}{\rho^2}\|\bar{\nu}_t - \nabla f(\tilde{x}_t)\|^2 = \|\tilde{d}_t\|^2 + \frac{1}{\rho^2}\|\bar{e}_t\|^2$$

Combine with Theorem B.16, we have:

$$\frac{1}{T}\sum_{t=0}^{T-1}\mathbb{E}[\mathcal{G}_t] \leq \left[\frac{96LK^{0.5}I^2}{T} + \frac{2L}{K^{2/3}T^{2/3}}\right](f(x_0) - f^*) + \left[\frac{72KI^4}{b_1 T} + \frac{3K^{0.5}I^2}{2b_1 K^{2/3}T^{2/3}}\right]\sigma^2$$

$$+ 192^2 \times \left(\frac{48K^{0.5}I^2}{T} + \frac{1}{K^{2/3}T^{2/3}}\right) \times \left(\frac{\sigma^2}{4b} + \frac{2\zeta^2}{21}\right)\log(T+1).$$

*Remark* B.17. For the measure $\mathcal{G}_t$, we discuss its intuition under both the unconstrained and constrained case. First, for unconstrained case, *i.e.* when $\mathcal{X} = R^d$, we have:

$$\|\nabla f(\tilde{x}_{\tau,i})\|/\|H_\tau\| = \|H_\tau \times H_\tau^{-1}\nabla f(\tilde{x}_{\tau,i})\|/\|H_\tau\| \leq \|H_\tau^{-1}\nabla f(\tilde{x}_{\tau,i})\|$$

$$= \|H_\tau^{-1}\nabla f(\tilde{x}_{\tau,i}) - H_\tau^{-1}\bar{\nu}_{\tau,i} + H_\tau^{-1}\bar{\nu}_{\tau,i}\| \leq \|H_\tau^{-1}\nabla f(\tilde{x}_{\tau,i}) - H_\tau^{-1}\bar{\nu}_{\tau,i}\| + \|H_\tau^{-1}\bar{\nu}_{\tau,i}\|$$

$$\leq \frac{1}{\rho}\|\bar{\nu}_{\tau,i} - \nabla f(\tilde{x}_{\tau,i})\| + \frac{1}{\eta_{\tau,i}}\|\tilde{x}_{\tau,i} - \tilde{x}_{\tau,i+1}\| \leq \sqrt{2}\sqrt{\mathcal{G}_{\tau,i}}/\rho$$

In the last inequality, we use Jensen inequality, and in the second last inequality, we use Assumption 5.4 and the fact that $\tilde{x}_{\tau,i+1} = x_{\tau,0} + H_\tau^{-1}\bar{z}_{\tau,i+1}$ and $\tilde{x}_{\tau,i} = x_{\tau,0} + H_\tau^{-1}\bar{z}_{\tau,i}$ and $\eta_{\tau,i}\bar{\nu}_{\tau,i} = \bar{z}_{\tau,i+1} - \bar{z}_{\tau,i}$ in the unconstrained case. In other words, we have $\|\nabla f(\tilde{x}_t)\|^2 \leq \frac{2\|H_\tau\|^2}{\rho^2}\mathcal{G}_\tau$. Note the coefficient of the right-side is an upper bound of the square condition number of $H_\tau$. It is common assumption in the analysis of adaptive gradient methods that $H_t$ has a finite condition

number Huang et al. (2021). In sum, the convergence of our measure $\mathcal{G}_t$ means the convergence to a first order stationary point in the unconstrained case.

Next, for the constrained case, our measure upper bounds the gradient mapping $\frac{1}{\eta_{\tau+1,i}}\|x_\tau - x^*_{\tau+1,i}\|$, $x^*_t$ is defined as follows:

$$x^*_{\tau+1,i} = \arg\min_{x\in\mathcal{X}}\{-\langle x, z^*_{\tau+1,i}\rangle + \frac{1}{2}(x-x_\tau)^T H_\tau (x-x_\tau)\}$$

where $z^*_{\tau+1,i} = \sum_{\ell=0}^{i} -\eta_\ell \nabla f(\tilde{x}_{\tau+1,i})$ is the accumulation of true gradient. Next follow Lemma B.6, we have:

$$\|x^*_{\tau+1,i} - \tilde{x}_{\tau+1,i}\| \le \frac{1}{\rho}\|z^*_{\tau+1,i} - \bar{z}_{\tau+1,i}\|$$

$$= \frac{1}{\rho}\|\sum_{l=0}^{i-1} -\eta_{\tau+1,\ell}(\nabla f(\tilde{x}_{\tau+1,\ell}) - \bar{\nu}_{\tau+1,\ell}))\| \overset{(a)}{\le} \sum_{l=0}^{i-1} \frac{\eta_{\tau+1,\ell}}{\rho}\|\nabla f(\tilde{x}_{\tau+1,\ell}) - \bar{\nu}_{\tau+1,\ell}\|$$

where inequality $(a)$ is due to the triangle inequality. Next we have:

$$\|x_\tau - x^*_{\tau+1,i}\| = \|x_\tau - \tilde{x}_{\tau+1,i} + \tilde{x}_{\tau+1,i} - x^*_{\tau+1,i}\| \le \|x_\tau - \tilde{x}_{\tau+1,i}\| + \|\tilde{x}_{\tau+1,i} - x^*_{\tau+1,i}\|$$

$$\le \|\sum_{l=0}^{i-1} \tilde{d}_{\tau+1,i}\| + \|\tilde{x}_{\tau+1,i} - x^*_{\tau+1,i}\| \le \sum_{l=0}^{i-1}\left(\|\tilde{d}_{\tau+1,\ell}\| + \frac{\eta_{\tau+1,\ell}}{\rho}\|\nabla f(\tilde{x}_{\tau+1,\ell}) - \bar{\nu}_{\tau+1,\ell}\|\right)$$

By Jensen inequality and the definition of the measure equation 9, we have

$$\|\tilde{d}_t\| + \frac{\eta_t}{\rho}\|\nabla f(\tilde{x}_t) - \bar{\nu}_t\| \le \frac{\sqrt{2}\eta_t}{\rho}\sqrt{\mathcal{G}_t},$$

So we have

$$\frac{1}{\eta_{\tau+1,i}}\|x_\tau - x^*_{\tau+1,i}\| \le \frac{\sqrt{2}}{\rho}\sum_{l=0}^{i-1}\frac{\eta_{\tau+1,l}}{\eta_{\tau+1,i}}\sqrt{\mathcal{G}_{\tau+1,l}} \le \frac{2\sqrt{2}}{\rho}\sum_{l=0}^{i-1}\sqrt{\mathcal{G}_{\tau+1,\ell}},$$

the last inequality is because of Eq. equation 17. In all, when the measure $\mathcal{G}_{\tau+1,\ell} \to 0$, the gradient mapping $\frac{1}{\eta_{\tau+1,i}}\|x_\tau - x^*_{\tau+1,i}\|$ converges to 0.

**Corollary B.18.** *With the hyper-parameters chosen as in Theorem B.16. Suppose we set $I = O((T/K^{3.5})^{1/6})$ and use sample minibatch of size $b_1 = O(K^{0.5}I^2)$ in the first step, Then we have:*

$$\mathbb{E}[\mathcal{G}_t] = O\left(\frac{f(x_0) - f^*}{K^{2/3}T^{2/3}}\right) + \tilde{O}\left(\frac{\sigma^2}{K^{2/3}T^{2/3}}\right) + \tilde{O}\left(\frac{\zeta^2}{K^{2/3}T^{2/3}}\right).$$

*and to reach an $\epsilon$-stationary point, we need to make $\tilde{O}(K^{-1}\epsilon^{-1.5})$ number of steps and need $\tilde{O}(K^{-0.25}\epsilon^{-1.25})$ number of communication rounds.*

*Proof.* It is straightforward to verify the expression for $\mathbb{E}[\mathcal{G}_t]$ in the corollary by applying Theorem B.16 and choosing $I$ and $b$ as corresponding values. As for the gradient and communication complexity of the algorithm. We have the following results: The number of total steps $T$ needed to achieve an $\epsilon$-stationary point, *i.e.* $\tilde{O}(\frac{1}{K^{2/3}T^{2/3}}) = \epsilon$, then the gradient complexity is $\mathcal{O}(K^{-1}\epsilon^{-3/2})$. Total rounds of communication steps to achieve an $\epsilon$-stationary point is $E = T/I$, as we have $I = O((T/K^{3.5})^{1/6})$, then $T/I = \tilde{O}(K^{7/12}T^{5/6})$. By the fact that $T = K^{-1}\epsilon^{-3/2}$, we have $E = O(K^{-1/4}\epsilon^{-5/4})$. $\qquad\square$