# OpenReview forum: "FedDA: Faster Adaptive Gradient Methods for Federated Constrained Optimization"
_ICLR.cc/2024/Conference — ICLR 2024 poster_

### Official Review · Reviewer_pEfb · 2023-10-27

**Soundness:** 3 good
**Presentation:** 3 good
**Contribution:** 3 good
**Rating:** 6
**Confidence:** 4

**Summary:**

In this work, the authors propose FedDA, an adaptive gradient method for constrained optimization in the Federated Learning context. The proposed method builds on top of previously released approaches in local adaptive gradients, and Federated composite optimization. The authors make their claims clear with both theoretical and experimental results (under homogeneous and heterogeneous data distributions). The main motivation behind their approach is to accelerate federated constrained optimization, by adopting the Mirror Descent view of momentum-based acceleration methods.

**Strengths:**

1- The method FedDA is general and can incorporate several adaptive gradient methods.

2- FedDA-MVR, the momentum-based variance-reduction gradient estimation, achieves the optimal  iteration complexity rates of non-convex stochastic optimization, without bounded gradient assumption.

3- FedDA performs better or on par with existing methods, on constrained biomarker identification tasks. FedDA-MVR performs well on image classification tasks, even on the challenging non-i.i.d. setting (Fig.12 in appendix).

Overall, the authors provide a clear overview of dual and/or adaptive Federated methods both theoretically (Table.1) and numerically (>5 concurrent methods tested in Fig.3).

**Weaknesses:**

1- $\tilde{x}_t$ in equation.9 is not defined in the main text. In particular, how do you compute the L2 distance in between $x_t$ and $\tilde{x}_t$? Your random variable $\mathcal{G}_t$ gives an upper-bound (rk.B.17) of L2-norm of the gradient on $\tilde{x}$ in the unconstrained case. Can we obtain the same for $x$? If yes, is the convergence rate to first order stationary point the same ?

2- I understand fine-tuning NNs on large image classification tasks can be time consuming. However, did you carefully tune the hyper-parameters for FedAvg method? FedAvg is known to perform well under the unconstrained homogeneous setting, but is below $45$% of test accuracy in your experiment on CIFAR-10 (homogeneous; Fig.3). Maybe playing on the local step value $I$, or increasing the batch size (fixed at $16$ in your experiments) can give better gradients estimates, and better performances for FedAvg.

3- Some typos about referencing your algorithms can be confusing (see details in the Questions section below).

**Questions:**

1- At the end of page.5, I think there is a typo about lines 9, 10, 11 of your Algo.1. In particular there is no line.11 in your Algo.1.

2- Minor: Please check your references before updating the final version. For instance the MiME paper is cited twice (on page.11 and on page.12). There is also a typo at the beginning of Section.6.1: PATHMN(I)ST.

---

> ### Author Response · Authors · 2023-11-20
>
> Thanks for your insightful review! Below is our response to your concerns:
>
> **W1**: *$\tilde{x}\_t$ in equation.9 is not defined in the main text. In particular, how do you compute......*
>
> **A**: In our theoretical analysis (Section 5.2), we define the global steps $t = \tau I + i$ for ease of analysis. Note that $\tau$ denotes $\tau_{th}$ global round, $I$ is the number of local steps, $i$ denotes the $i_{th}$ local step. As a result, $t$ denotes the $t_{th}$ overall step during training. In Algorithm 1, we have **the global primal state at the $\tau_{th}$ global round to be $x_{\tau}$**. Furthermore, we define **the virtual global state at the $i_{th}$ step of $\tau$ global round as $\tilde{x}_{\tau, i}$** (defined in eq. 12 and eq. 13 in Appendix B.2 of the manuscript).  It is straightforward to verify that we have $\tilde{x}\_{\tau, I}= x_{\tau+1}$. Meanwhile, by the definition of $t = \tau I + i$, we have $\tilde{x}\_t = \tilde{x}\_{\tau, i}$. As a result, **we have $x\_{\tau} = \tilde{x}\_{t}$, for $ t = \tau I$ and $\tau \in [E]$**. In summary, $\tilde{x}\_{t}, t\in [T]$ is the full sequence of global primal states, and $x_{\tau}, \tau \in [E]$ is the global primal state at each server aggregation step. The upper bound result in Remark B.17 also holds for $\{x_{\tau}\}$, and the convergence of $\{\tilde{x}_t\}$ also leads to the same convergence rate of $\{x\_{\tau}\}$.
>
> **W2**: *However, did you carefully tune the hyper-parameters for FedAvg method? FedAvg is known to perform well under the unconstrained homogeneous setting, but is below......*
>
> **A**: Thanks for pointing out this. We carefully check the code implementation for FedAvg and find that, for CIFAR10 (Homogeneous) and FEMNIST, we do not average local states correctly every $I$ steps. As a result, the model overfits training data severely and gets low test accuracy (below 45\%). In the updated version, we have updated Figure 3 to include correct results. In fact, FedAvg reaches test accuracy of 59.84\% after 4000 global rounds (Our FedDA still outperforms FedAvg after the correction). In our experiments, we use the same local steps and batch-size for all methods for fair comparison. However, the number of local steps and batch-size indeed affects the convergence. Below we show some ablation study results for FedAvg over the CIFAR10 (Homegeneous) dataset:
>
> **Table 1**: Test Accuracy for FedAvg under different number of local steps $I$
>
> |  I\Global Steps   | 1     | 1000  | 2000  | 3000  | 4000  |
> |-----|-------|-------|-------|-------|-------|
> | 5   | 12.50 | 46.09 | 53.38 | 57.43 | 59.84 |
> | 10  | 18.75 | 53.95 | 60.45 | 63.25 | 64.46 |
> | 20  | 12.50 | 60.48 | 64.69 | 65.81 | 66.43 |
> | 50  | 12.50 | 65.31 | 67.35 | 67.69 | 68.05 |
>
> **Table 2**: Test Accuracy for FedAvg under different batchsizes
>
> | Batchsize\Global Steps    | 1     | 1000  | 2000  | 3000  | 4000  |
> |-----|-------|-------|-------|-------|-------|
> | 16  | 12.50 | 46.09 | 53.38 | 57.43 | 59.84 |
> | 32  | 25.00 | 48.19 | 54.75 | 58.30 | 60.60 |
> | 64  | 7.81  | 48.61 | 55.59 | 59.20 | 61.24 |
> | 128 | 11.72 | 49.59 | 56.38 | 59.79 | 61.64 |
>
> As shown by Table 1 and Table 2, increasing local steps and batch-size both improve the test accuracy of FedAvg, but the number of local steps $I$ has a larger effect to test accuracy than the batch-size.
>
> **W3**: Some typos about referencing your algorithms can be confusing (see details in the Questions section below).
>
> **A3**: In the end of page 5, both $\nu\_{\tau}$ and $H\_{\tau}$ are updated at Line 9 of Algorithm 1. We also corrected the double reference issue to the MIME paper and the other typo you mentioned in the updated version. We will also do a careful proofreading in the final version.

---

> > ### Author Response · Authors · 2023-11-23
> >
> > Dear Reviewer,
> >
> > Thanks for your review again! If you have any further questions or concerns, please do not hesitate to let us know.
> >
> > Authors

---

### Official Review · Reviewer_hZzf · 2023-10-28

**Soundness:** 3 good
**Presentation:** 1 poor
**Contribution:** 3 good
**Rating:** 6
**Confidence:** 3

**Summary:**

This paper introduces a novel approach to addressing the problem of constrained federated learning by leveraging adaptive gradient methods. In conventional, non-federated contexts, various techniques and algorithms are available for finding local minimizers of functions, with adaptive gradient methods being particularly effective. The authors extend this idea into the federated learning framework by creating a general adaptive gradient framework.

This framework includes multiple adaptive gradient methods that rely on a restarted dual averaging technique. The authors begin by adopting a mirror descent perspective of adaptive gradient methods. The central concept in this paper is that if all clients share a common mirror map, they will operate in the same dual space. Consequently, the server can aggregate the local dual states of different clients effectively.

The paper provides a theoretical analysis of the convergence rate for a specific instantiation of this framework, which employs a momentum-based variance reduction technique. To validate their approach, the authors conduct empirical experiments on various datasets, demonstrating its effectiveness.

**Strengths:**

- The authors propose an algorithm that solves the under-explored field of constrained federated learning with optimal convergence rates for the constrained setting (up to my knowledge).
- The algorithm proposed by the authors is projection-free which is remarkable in a constrained optimization problem.

**Weaknesses:**

- I found the paper to be poorly written and presented.
- While I did not found the proposed method to be highly competitive in the unconstrained setting, I believe it holds significant promise in the constrained setting. In my view, the authors should have emphasized the motivation and results from this perspective, rather than aiming to compete with potentially more efficient existing methods.
- This algorithm is more computationally demanding compared to other algorithms. The added computational cost arises from computing an argmin at each iteration. It remains unclear whether, with this additional step, the proposed algorithm maintains the same level of computational efficiency as its competitors and achieves similar theoretical and experimental results within the same computational budget.
- While the assumptions made by the authors are common in the analysis of adaptive gradient methods, I find some of them to be rather restrictive, particularly Assumptions 5.1 and 5.4. In light of this observation, the comparison with non-adaptive methods seems somewhat unfair.

**Questions:**

- Can we check that assumption 5.4 really holds for the provided update of the adaptive matrix in the case of the momentum-based variance reduction instantiation?

---

> ### Author Response · Authors · 2023-11-20
>
> Thanks for your insightful review! We will polish the writing and presentation of our paper in the final version, and below is our response to your concerns:
>
> **W1**: *In my view, the authors should have emphasized the motivation and results from this perspective, rather than aiming to compete with potentially more efficient existing methods.*
>
> **A**: We agree with the reviewer that the most interesting application of our FedDA framework lies in the constrained FL. Indeed, in the introduction, we motivate FedDA through the FL healthcare applications, especially the biomarker identifications tasks which often require constrained formulations. Next, in the numerical experiments section, we verify our FedDA through two fed-biomarker identification tasks: the colorrectal cancer prediction task and the splice site detection task. However, our FedDA can also be applied to the unconstrained setting. For completeness, we compare the convergence rate of FedDA with other widely used FL methods in Table 1 and also perform unconstrained image classification over CIFAR-10 and FEMNIST in Section 6.3. In the final version, we will add more discussion about the application of FedDA to the constrained setting.
>
> **W2**: *This algorithm is more computationally demanding compared to other algorithms......*
>
> **A**: As mentioned by the reviewer, we have an argmin operation in Line 8 of Algorithm 1 (similarly, Line 5 in Algorithm 2), but **this is indeed a generalized adaptive update and is not an additional step**. In fact, we solve this argmin through two steps. Firstly, we get an explicit solution of the quadratic objective, denoted as $x_{\tau+\frac{1}{2}} = x_{\tau} - H_{\tau}^{-1}z_{\tau+1}$. This can be viewed as a generalized adaptive gradient update and incorporates many existing adaptive gradient methods. For example, Adam uses the exponential moving average of gradient square for adaptive gradients, while in our case, $H_{\tau}$ is a diagonal matrix with the diagonal elements containing this moving average. Next, we perform a projection step: $x_{\tau+1} = Proj_{\mathcal{X}}(x_{\tau+\frac{1}{2}} )$. This step is necessary to make the solution feasible. In fact, we have explicit solutions for many common projectors. For example, we have soft-thresholding operator if the constraint set is an $L_1$ ball. In summary, compared to existing adaptive gradient methods developed for unconstrained setting, the argmin operation does not leads to extra computational cost (except an essential projection operation).
>
> **W3**: *particularly Assumptions 5.1 and 5.4. In light of this observation, the comparison with non-adaptive methods seems somewhat unfair.*
>
> **A**: In fact, Assumption 5.1 is used for measuring the heterogeneity effects of FL devices and is not particular related to adaptive gradient methods. In Theorem 3.2 of [1], authors also analyze the convergence rate of FedAvg based on this heterogeneity assumption.
>
> Assumption 5.4 relates to adaptive learning rates and is used to guarantee that the adaptive matrix is positive definite. In fact, to satisfy Assumption 5.4, we can add a small number $\epsilon$ over $H_{\tau}$ (e.g. eq.7 and eq.8 in the manuscript). Take Adam as an example (eq.7), since we maintain the moving average of the square of the gradient, the adaptive learning rates are guaranteed to be non-negative, then adding an additional small number of $\epsilon$ makes sure that the adaptive matrix $\epsilon$-positive definite.
>
> In experiments, we set $\epsilon = 0.01$ for FedDA-MVR (which uses Adam-style adaptive gradients or eq. 7 in the manuscript), so we have the $H_{\tau}$ to be 0.01-positive definite. (This also answers the question raised by the reviewer)
>
>
> **References**
>
> *[1]. Khanduri, Prashant, et al. "Stem: A stochastic two-sided momentum algorithm achieving near-optimal sample and communication complexities for federated learning." Advances in Neural Information Processing Systems 34 (2021): 6050-6061.*

---

> > ### Author Response · Authors · 2023-11-23
> >
> > Dear Reviewer,
> >
> > Thanks for your review again! If you have any further questions or concerns, please do not hesitate to let us know.
> >
> > Authors

---

### Official Review · Reviewer_VFUm · 2023-11-05

**Soundness:** 3 good
**Presentation:** 2 fair
**Contribution:** 3 good
**Rating:** 6
**Confidence:** 3

**Summary:**

The authors propose an adaptive gradient approach for federated constraint optimization. While existing schemes have focused separately on federated optimization and adaptive gradient approaches for the centralized setting, they combine these two lines of research. Specifically, they derive order results for the convergence and the communication complexity of their proposed scheme and perform numerical experiments for a variety of homogenous and non-homogenous datasets.

**Strengths:**

The authors have filled a missing gap in proposing adaptive gradient schemes for FL, which did not seem to exist in the open literature before. This is useful for constraint optimization in a federated setting, for example for regularization.

**Weaknesses:**

To my opinion, there are two major weaknesses. First, I think that the underlying idea of combining dual averaging with federated constraint optimization is fairly straightforward. The merit of the work is rather in the theoretical order complexity results.

Second, the paper lacks important information, and thus it is difficult to evaluate their results, specifically with respect to the adaptive gradient results. Specifically, they state on page 5 that the mapping matrix H is updated in line 11 of Algorithm 1, however, line 11 is not present in that algorithm.

**Questions:**

What is the comparison of the sample and communication complexity to other non-adaptive schemes? A little table containing these results would be useful.

In Figs. 1 and 2, the third plot from the right is not explained in the text. Also, here, essentially the gain of FedDA is due to the addition of a regularization constraint (Lasso). Is there also a gain over benchmark schemes (as FedAvg) for settings where these schemes are not overfitting and why?

---

> ### Author Response · Authors · 2023-11-20
>
> Thanks for your insightful review! Below is our response to your concerns:
>
> **W1**: *the underlying idea of combining dual averaging with federated constraint optimization ...... the work is rather in the theoretical order complexity results.*
>
> **A**: A main challenge of applying adaptive gradients to federated constrained optimization lies in how to exchange information among clients in presence of complicated constraints condition.  FedDA alleviates this challenge by **disentangling** the constraint condition (in the primal space) from adaptive gradient updates (in the dual space). More specifically, FedDA adopts the mirror descent view of adaptive gradients and aggregates gradient updates in the dual space; By aligning dual spaces among clients (fix the adaptive matrix during local updates), FedDA facilitates the **effective information exchange among clients** through averaging local dual states. Finally, we agree with the reviewer that theoretical analysis of FedDA is challenging, particular due to the non-linear mapping between primal and dual spaces. However, we successfully show FedDA-MVR (an instantiation of FedDA) obtains the optimal iteration complexity.
>
> **W2**: *they state on page 5 that the mapping matrix H is updated in line 11 of Algorithm 1, however, line 11 is not present in that algorithm.*
>
> **A**: In Algorithm 1, the update of the adaptive matrix H is in Line 9. We have corrected this typo in the updated version.
>
> **Q1**: *What is the comparison of the sample and communication complexity to other non-adaptive schemes? A little table containing these results would be useful.*
>
> **A**: We agree with the reviewer that it is beneficial to compare with other non-adaptive algorithms. Indeed, we compare the sample and communication complexity of our algorithm and other baselines in Table 1 of the manuscript. In Table 1, *Gc* represents sample complexity and  *Cc* represents the communication complexity of algorithms.
>
> **Q2**: *In Figs. 1 and 2, the third plot from the right is not explained in the text ......*
>
> **A**: The third plot of Figure 1 shows the density of model parameter, *i.e.* the percentage of non-zero elements in model parameter, and this is used to measure the sparsity of the learned model. The third plot of Figure 2 shows the Pearson
> correlation between the binary random variable of the true class membership and the binary random variable of the predicted class membership, and this is used to measure the test performance of the learned model.
>
> In Figure 1 and Figure 2, we also include results of FedDA without regularization constraints and FedDA without constraints also outperforms the baseline method FedAvg. Furthermore, in Section 6.3, we run the classification task over the CIFAR-10 and FEMNIST datasets. In this task, we don't need the constraint to avoid overfitting, however, FedDA still outperforms various baselines including FedAvg. Note that FedAvg is based on the vanilla stochastic gradient descent, while FedDA incorporates the momentum and adaptive gradients techniques. Thus, FedDA can outperforms FedAvg.

---

> > ### Author Response · Authors · 2023-11-23
> >
> > Dear Reviewer,
> >
> > Thanks for your review again! If you have any further questions or concerns, please do not hesitate to let us know.
> >
> > Authors

---

### Meta-Review · Area_Chair_8mQB · 2023-12-04

**Metareview:**

In this paper, the authors proposed a general adaptive gradient approach to solve the constrained federated optimization problems, several adaptive gradient methods can be covered in this approach.    $O(K{-1}\epsilon^{-1.5})$ sample complexity and $O(K^{-0.25}\epsilon^{-1.25})$ communication complexity are derived, and numerical experiments are conducted. The major complaint from reviewers lies in the novelty of the result, i.e., the proposed method is a straight combination of various existing tricks. It also has some merit in handling the constraints in the federated learning. Overall, we suggest a accept to this paper.

**Justification For Why Not Higher Score:**

The proposed method is a straight combination of various existing tricks.

**Justification For Why Not Lower Score:**

The paper has some merit in handling the constraints in the federated learning.

---

### Decision · Program_Chairs · 2024-01-16

Accept (poster)